# Deep multiple instance learning versus conventional deep single instance learning for interpretable oral cancer detection

**Nadezhda Koriakina**[1]*, **Nataša Sladoje**[1], **Vladimir Bašić**[2,3], **Joakim Lindblad**[1]

**1** Centre for Image Analysis, Department of Information Technology, Uppsala University, Uppsala, Sweden, **2** Department of Natural Science and Biomedicine, School of Health and Welfare, Jönköping University, Jönköping, Sweden, **3** Clinical Research Center Dalarna, Uppsala University, Falun, Sweden

* nadezhda.koriakina@it.uu.se

**Data Availability Statement:** PAP-QMNIST data are available from the Zenodo repository, 10.5281/zenodo.7020311. The oral cancer data cannot be shared publicly because of ethical permit

## Abstract

The current medical standard for setting an oral cancer (OC) diagnosis is histological examination of a tissue sample taken from the oral cavity. This process is time-consuming and more invasive than an alternative approach of acquiring a brush sample followed by cytological analysis. Using a microscope, skilled cytotechnologists are able to detect changes due to malignancy; however, introducing this approach into clinical routine is associated with challenges such as a lack of resources and experts. To design a trustworthy OC detection system that can assist cytotechnologists, we are interested in deep learning based methods that can reliably detect cancer, given only per-patient labels (thereby minimizing annotation bias), and also provide information regarding which cells are most relevant for the diagnosis (thereby enabling supervision and understanding). In this study, we perform a comparison of two approaches suitable for OC detection and interpretation: (i) conventional single instance learning (SIL) approach and (ii) a modern multiple instance learning (MIL) method. To facilitate systematic evaluation of the considered approaches, we, in addition to a real OC dataset with patient-level ground truth annotations, also introduce a synthetic dataset—PAP-QMNIST. This dataset shares several properties of OC data, such as image size and large and varied number of instances per bag, and may therefore act as a proxy model of a real OC dataset, while, in contrast to OC data, it offers reliable per-instance ground truth, as defined by design. PAP-QMNIST has the additional advantage of being visually interpretable for non-experts, which simplifies analysis of the behavior of methods. For both OC and PAP-QMNIST data, we evaluate performance of the methods utilizing three different neural network architectures. Our study indicates, somewhat surprisingly, that on both synthetic and real data, the performance of the SIL approach is better or equal to the performance of the MIL approach. Visual examination by cytotechnologist indicates that the methods manage to identify cells which deviate from normality, including malignant cells as well as those suspicious for dysplasia. We share the code as open source.

restrictions. If you wish to obtain access to the oral cancer datasets, you must first obtain permission from the Swedish Ethical Review Authority. Their contact information is as follows: Telephone Number: +46-10-475 08 00, E-mail: registrator@etikprovning.se. The values used to build graphs, the values behind the means, standard deviations, and other measures reported, as well as the weights for neural network models trained on PAP-QMNIST and oral cancer data, are provided in Zenodo repository XXX. (we insert the link upon acceptance).

**Funding:** This work is supported by: Sweden's Innovation Agency (VINNOVA) https://www.vinnova.se/en/apply-for-funding/funded-projects/, grants 2017-02447, (J.L.), 2021-01420 (J.L.), and 2020-03611 (J.L.), the Swedish Research Council https://www.vr.se/english/swecris.html#/, grant 2017-04385 (J.L.) and 2022-03580_VR (N.S.), and Cancerfonden https://www.cancerfonden.se/forskning, project number 22 2353 Pj (J.L.) and project number 22 2357 Pj (N.S.). The funders had no role in study design, data collection and analysis, decision to publish, or preparation of the manuscript. There was no additional external funding received for this study.

**Competing interests:** The authors have declared that no competing interests exist.

## Introduction

Cancers of the oral cavity and oropharynx are on the list of the most common malignancies in the world. Early detection of cancer is highly desirable as it is a premise of successful treatment. The current gold standard for confirming a diagnosis is histological examination of a tissue biopsy sample, which is time-consuming and painful for the patient. An alternative is to develop painless methods which rely on taking brush samples from the oral cavity of the patients and subsequently analyze the collected and suitably prepared cytological data. A trained cytotechnologist is able to detect abnormalities in samples acquired from patients with malignancy by carefully examining the cells with a microscope. However, this is a very difficult task; a recent study indicates sensitivity and specificity reached by human experts *on the patient-level diagnosis* being not more than 80% and 86%, respectively, in oral cancer cytology screening [1]. Steps towards improving diagnostic accuracy, but also steps towards improved understanding of the malignancy, its causes and consequences, are therefore very valuable.

Automated analysis of cytological data can increase the efficiency of the process to a level that opens possibilities for population-wide screening, towards early cancer detection. Deep learning (DL) based image classification methods have shown the ability to detect differences between malignant versus healthy samples, without the need for very difficult and time-consuming labeling of each individual cell [2]. Systems based on such methods for oral cancer (OC) detection could assist cytotechnologists, given that the method is reliable enough. For successful adoption of DL-supported cancer detection in healthcare, it is not sufficient for a classifier to provide just an answer 'yes' or 'no'. In order to trust the classifier's predictions, a human expert needs information about *why* the system reaches a certain decision [3]. An example of such explanatory information is a provided set of cells identified by the system as malignant, leading to the patient-level decision of detected cancer. However, (even) to evaluate such information becomes a challenging task when only per-patient, and not per-cell, labels are available, as is typically the case for cytology OC data. While the relatively few, subjectively judged, per-cell annotations provided by a cytotechnologist are not sufficient for training, thorough evaluation, nor reliable interpretation of the DL classification methods, they are still useful for observing similarities and differences between human and machine findings.

Aiming towards efficient and trustworthy cytology-based OC detection, we are interested in DL-based methods that are able to learn from only patient-level labels and demonstrate good patient-level performance while also providing information about the importance of individual cells for a diagnosis, thereby facilitating human interpretation and supervision of the decision made by the DL method.

To tackle this challenge, we are following emerging approaches taken in Whole Slide Image (WSI) analysis for (bio)medical applications [4], where labeling of all parts of a WSI is not feasible, and only weak labels are available. More precisely, we are addressing OC detection as a weakly supervised learning problem. Due to the large size of WSIs, the solutions typically involve dividing each WSI into many smaller patches, which are further processed. We utilize this idea too, while still striving to preserve and use holistic, slide-level information. Publicly available datasets for computational cytology do not meet the criteria of preserving slide-level information and are composed of preselected cells or tiles. Other (non-public) cytology datasets that are comprised of WSIs are most often suitable for detection of cervical, not oral cancer [5]. For an in-house collected OC dataset, we choose an automated approach which allows to consider all the cells in the WSIs. This is in line with the clinical routine, where cytotechnologists typically look at a whole WSI to find a few abnormal cells.

Multiple instance learning (MIL) is a type of weakly supervised learning which can offer methods suitable for OC detection [6]. While *bags* are composed of (multiple) instances, the

core idea of MIL is to treat the bags as individual objects, each with a well-defined label. When MIL is used for binary classification, the aim is to classify negative bags against positive bags; a bag is classified as positive if it contains at least one positive (key) instance, whereas a negative bag does not include any key instances.

WSI analysis with patient-level labels can be seen as an example of MIL, with a set of labeled bags (patient samples), such that a bag is labeled positive if at least one instance (one cell) in it is positive (malignant). An alternative to MIL is a conventional deep single instance learning (SIL) approach. This approach can be seen as a differently formulated weakly supervised learning task, where, in the absence of per-cell annotations, each cell is given a label corresponding to the label of the sample/patient it comes from. Such a weak, patient-level label is an unreliable indicator of cell-level malignancy because a sample acquired from a patient with a malignancy may include very many normal (non-malignant) cells.

In this study, we compare two methods, one MIL-based and one conventional SIL-based approach (see an overview of the methodology in Fig 1). The methods are conceptually different but are both based on deep convolutional neural networks (CNN) and can be trained end-to-end using only weak patient-level labels. Our main interest is to: (i) compare the performance of the methods, on the per-patient level, as well as on the per-cell level prediction; and (ii) explore the decision-making strategies of both approaches by evaluating the networks' ability to identify the key instances, i.e., the diagnostically most relevant cells.

To be able to reliably evaluate the observed methods, we have created PAP-QMNIST, a synthetic dataset that mimics several properties of WSI OC data relevant for training a CNN in both MIL and SIL regimes, such as cell image size, color distribution, arbitrary rotation of cells, presence of blur and noise, large and varied number of instances per bag. PAP-QMNIST consists of images of digits that are altered by image transformations, i.e., rotation by a random angle, colorization, etc. We create a synthetic ground truth (GT) at the instance level by choosing a certain digit to represent key instances in positive bags of PAP-QMNIST. In addition, it is possible to vary the percentage of key instances in positive bags of PAP-QMNIST. We note that creating a (more realistic) synthetic dataset composed of images mimicking cells with reliable GT annotation at the instance (cell) level is not feasible because properties that distinguish malignant versus healthy cells are not fully known. A main advantage of PAP-QMNIST is that it offers access to reliable synthetic GT annotation at the instance level in combination with being visually interpretable for non-experts. The PAP-QMNIST dataset allows us to analyze the behavior of methods, e.g., when the methods start to fail or demonstrate dissimilar results. Such analysis is much more challenging on real data with limited GT and complex interpretation. We use the PAP-QMNIST dataset to perform a thorough bag (per-patient) and instance (per-cell) level evaluation of end-to-end DL approaches applied to weakly labeled data that has not been previously demonstrated. We envision that the PAP-QMNIST dataset may be useful for similar studies by other researchers in the field.

The main contributions of our work are: (i) We introduce a new dataset, PAP-QMNIST, as a proxy model with similar distribution as the cytological data, with a well defined and visually assessable GT; (ii) We use PAP-QMNIST to evaluate the performance of two different approaches for weakly supervised malignancy detection; (iii) We compare the two observed approaches on real medical cytological OC WSI data, focusing on both per-patient and per-cell performance, and analyze to what extent the findings on the synthetic data transfer to the real cytology data.

The main findings of our work are: (a) We observe that it is possible to detect malignancy (key instances) in both synthetic and real data using both observed approaches, and we observe similar performance at the patient (bag) level classification for both approaches. We also observe that the SIL approach outperforms the MIL approach in the detection of malignant

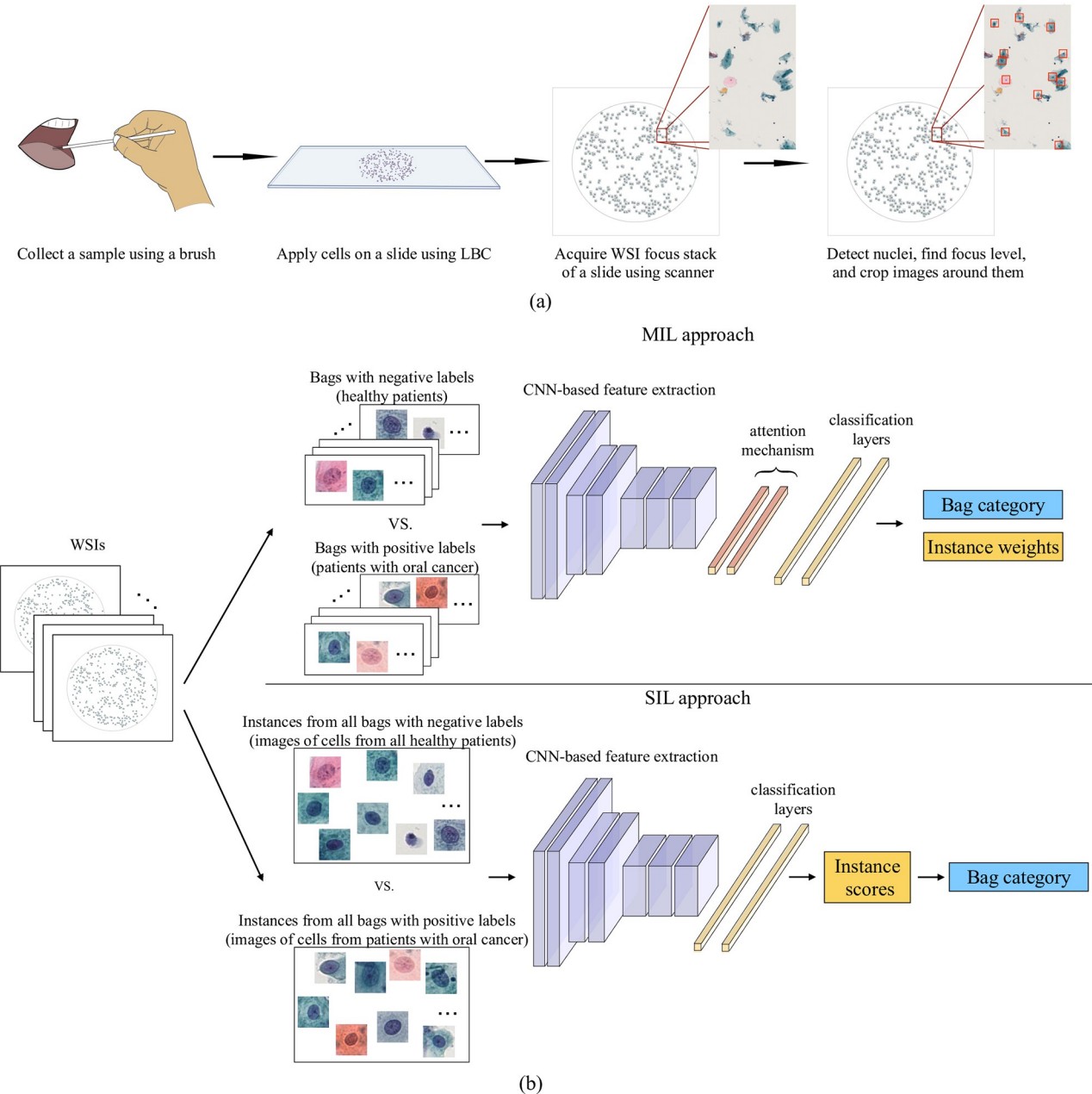

**Fig 1. Overview of the methodology.** (a) Workflow for acquisition of cytology images and (b) schematic diagrams of MIL and SIL approaches. Both approaches are trained end-to-end and provide bag (patient) category and interpretability at the instance (cell) level. We consider the MIL approach with the attention mechanism for obtaining instance weights to interpret the provided bag-level decision. The SIL approach provides instance scores, according to which we find the bag category.

cells (key instances); (b) By analyzing the key instances detected in PAP-QMNIST data, we reveal the tendency of the MIL approach to focus on instances with certain features not relevant for the classification task.

To facilitate reproducibility, we share the complete implementation and evaluation framework as open source.

https://github.com/MIDA-group/OralCancerMILvsSIL.

## Background and related work

With our main interest in the development of an efficient and reliable automated system for OC detection, we investigate existing weakly supervised methods, which are applicable also in cases with a limited amount of labeled data. This is stated as one of the remaining challenges in computational pathology by Lu et al. [7].

Our previous work [2] indicates that it is possible to distinguish cells from healthy patients and cells coming from OC patients, using only per-patient annotations. This method is based on single instance-level training of deep CNN and we use it as a reference method in our study. The same strategy is evaluated in recent studies, for example, in Li et al. [8], where it is referred to as "patch-based without considering MIL". In our current study, we explore this approach further, to understand whether it can help to find cells that are highly relevant for OC diagnosis.

There are other methods that can be relevant for our problem. They lie on the intersection of two research directions: (i) weak supervision, due to the absence of cell-level annotations and (ii) interpretability, providing cell-level information to reach trustworthy solutions and reliable utilization. The approaches that can meet both these requests and at the same time perform well for WSI analysis are: attention-based deep MIL (ABMIL) [9], a method by Campanellaet al. [10], dual-stream MIL (DSMIL) [8], clustering-constrained-attention MIL (CLAM) [7], transformer-based MIL (TransMIL) [11], and a more recent end-to-end weakly supervised learning framework (EWSLF) [12]The work on ABMIL proposes to incorporate attention mechanisms (and by that interpretability) to MIL, to provide insight into the contribution of each instance. The method by Campanella et al. involves aggregation for MIL based on recurrent neural network (RNN). DSMIL is a fusion of a novel non-local attention-based pooling operator with self-supervised contrastive learning and a multiscale pyramidal scheme to extract representations. CLAM is an attention-based method with instance-level clustering. EWSLF [12] is an end-to-end method with cluster-based sampling strategy and multi-branch attention mechanism. TransMIL is designed using Transformer [13] and considers correlation among instances in a bag.

With our current focus on evaluating the feasibility of using weakly supervised learning methods on in-house OC data from a low number of patients, with a large number of cells for some patients, and unbalanced number of cells among patients, not all above mentioned methods are equally applicable. Some of the algorithms are partly tailored specifically for histology, and not cytology data analysis; they consider spatial relation among instances (cells), which is not relevant for the rather randomly distributed cells on cytology WSIs. Such are the introduction of RNN for aggregation in Campanella et al. [10], the multiscale approach in Li et al. [8], and the pyramid position encoding generator in TransMIL [11].

In this study, along with conventional (single instance) DL approach involving training per patch using weak labels, we consider a frequently used representative of MIL methods— ABMIL—which is observed to perform well on similar tasks. This method both demonstrates good performance at a bag level and also offers interpretability at an instance level. ABMIL is an end-to-end method, and so is the SIL approach we consider; we believe that end-to-end approaches can learn features that represent data better than features obtained in a non-end-to-end manner (e.g., engineered or pretrained). However, ABMIL cannot be applied directly on bags with a very large number of instances (such as, e.g., WSI in cytology, which may have more than 100,000 cells per slide) due to GPU memory constraints. Other MIL approaches tackle this problem using, for example, self-supervision [8], selection of instances in a bag during training [10] or features learned by models pretrained on ImageNet [7, 11]. In a previous study [14], we observed that ABMIL can work well with within-bag sampling, benefiting from

end-to-end learning, while overcoming memory requirements imposed by WSI. Similarly to ABMIL with sampling, EWSLF [12] is trained end-to-end and offers reduced computational cost of handling large WSI data. However, it is designed for cancer subtype classification and would not demonstrate its full advantage in classifying positive versus negative classes. Further, the inference scheme in [14] takes into account sampling from large bags at test time unlike the inference in [12]. Therefore, in this study, we evaluate ABMIL with sampling, following [14], on OC data and synthetic PAP-QMNIST data.

## Comparative evaluation

### Methods

**Ethical approval.** The study was performed in compliance with the Declaration of Helsinki and approved by the Ethical Review Board Stockholm Sweden (2015–1213-31 and 2019–00349). Informed written consent was obtained from all participants. The study does not involve minors. Data were accessed for research purposes during 2020 and 2021.

**Considered single instance learning—SIL—method.** We follow the weakly supervised approach proposed in Lu et al. [2], i.e., we use CNN to classify cells (instances), labeled as the patient (bag) they come from, into two classes, healthy and cancer. To select a model for inference, we observe the performance on a validation set; we choose to use F1 score, suitable in the presence of the class imbalance in terms of the number of instances. We predict the bag label by aggregating the instance predictions for each bag; if a test bag in fold $f$ contains a percentage of positively classified instances which is higher than a threshold $t_f$ computed using train and validation bags in fold $f$, then the test bag is assigned a positive label.

**Considered multiple instance learning—MIL—Method.** We use a version of ABMIL, Ilse et al. [9], with modifications as proposed in Koriakina et al. [14], i.e., instead of using original bags (non-feasible for the OC dataset due to the very large bags, containing all cells in a WSI), we use smaller mini-bags sampled with replacement from the original bags. Sampling with replacement is applied during training and inference time to introduce diverse combinations of instances from original bags. Our previous study indicated an increase in Area under the ROC Curve (AUC) at the instance level for smaller mini-bags, as long as there is a sufficient (dataset dependent) number of key instances per mini-bag to be detected [14]. Due to the balanced number of bags (patients) for the two classes in the in-house data, we may directly rely on bag classification accuracy for selecting a model for inference.

### Data

**Oral cancer dataset.** The OC Dataset consists of liquid-based (LBC) prepared PAP-stained slides of brush-sampled cells from the oral cavity of 24 patients. Among 24 patients in the OC Dataset, there are 12 patients with histologically confirmed malignancy and 12 healthy patients. Slides were imaged using a NanoZoomer S60 slide scanner, 40×, 0.75 NA objective, at 11 z-offsets (stepping 0.4 μm) providing RGB WSIs of size 103, 936 × 107, 520 × 3, 0.23 μm/pixel.

The method described in Lu et al. [2] was used to detect cell nuclei for which $80 \times 80$ pixel regions were cut out, each with one centered nucleus in focus. This provides in total 196, 243 cells from healthy and 111, 596 cells from OC patients. Example images are shown in Fig 2(a) and 2(b). The patient-level diagnoses are set based on histological tissue sample analysis (used as GT), where "positive" labels are associated with patients with a detected malignancy. To assign a reliable cell-level annotation for every cell is not feasible; it is not only time-consuming but also a too difficult task to perform even by a highly skilled cytotechnologist.

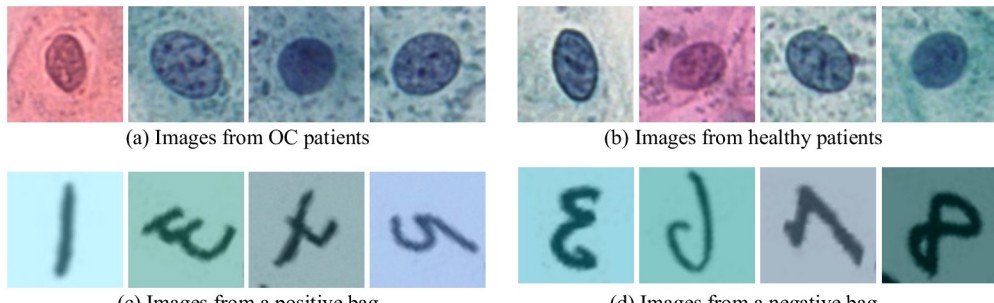

(a) Images from OC patients                    (b) Images from healthy patients

(c) Images from a positive bag                    (d) Images from a negative bag

**Fig 2. Examples of images from datasets.** Top: OC; Bottom: PAP-QMNIST. Images of the digit "4" constitute key instances in the PAP-QMNIST dataset and appear only in positive bags (an example is the third image in (c)), whereas images of other digits appear in both positive and negative bags.

**PAP-QMNIST dataset.**   The lack of reliable cell-level annotations in OC data makes it difficult to judge the outcome of experiments, e.g., whether the dataset is too challenging to be processed by a particular method or the parameters of the method are not tuned properly. To facilitate method development and evaluation, we introduce a new synthetic dataset, PAP-QMNIST (S1 Dataset), which is created with several properties of OC data, i.e., image size, color distribution, arbitrary object rotation, presence of blur and noise, large and varied number of instances per bag. In contrast to OC data, PAP-QMNIST has an advantage of having reliable (defined by design) GT at the instance level. To facilitate clear interpretation of our studies, we intentionally chose to utilize objects which are easy to recognize (digits) and where there is no doubt on which properties are important; the shape is the important property for the PAP-QMNIST classes, while color or rotation, for example, are not.

Previously, MNIST-bags [9], CIFAR10-bags [15], QMNIST-bags [14], and Imagenette-bags [14] have been used for development and evaluation of MIL approaches. These datasets are not well suited to serve as a proxy model of the in-house OC Dataset; they are not of appropriate size (they are too small) and do not provide enough images per bag (patient). PAP-QMNIST (see Fig 2(c) and 2(d)) is created taking these issues into account. We base PAP-QMNIST on the QMNIST dataset [16], for which the object (digit) is located in the central part of the image, similarly to (the detected and cut-out) nuclei in the in-house OC Dataset, and where the large number of images enables us to create a larger dataset than above mentioned existing datasets for MIL development.

We rescale original QMNIST images to the size of images in the OC Dataset using bilinear interpolation, add color, and augment this dataset by including images transformed by transformations expected in OC data to replicate the number of patients and number of images per patient in the OC Dataset. An example of transformations expected in OC data is a rotation by a random angle, since cells in the OC Dataset may appear in any rotation. A full list of transformations is presented in S1 Table. The colorization of QMNIST images is performed as follows: from a sample of 5280 images from the OC Dataset, we plot the histogram of each color channel and approximate them by a normal distribution for the green and blue channels and a skewed normal distribution for the red channel. The cells in the OC Dataset have predominantly blue/turquoise background, therefore, we sample color values from the distributions for each color channel which, in combination with the other two channels, give colorization visually close to blue/turquoise color. We achieve such colorization by avoiding sampling from the tails of the distributions. The same colorization approach is applied to positive and negative classes and the colorization does not affect the class label. Therefore, color is by explicit and

intentional design not a relevant feature for PAP-QMNIST. This *a priori* knowledge is useful when evaluating the decision-making of the trained networks; if color appears as an important feature for a certain class of PAP-QMNIST, it is an indication that the network probably has over-fitted the training data. Colorization cannot be omitted for the PAP-QMNIST creation because color, in general, may carry information about OC data and the considered classification approaches need to be able to handle colors (and learn if it matters for classification, or not).

The original QMNIST dataset is composed of train and test sets, each with 60k images. We randomly extract 20% of the original QMNIST test set to form a validation set while the remaining 80% are kept as a test set. PAP-QMNIST train, validation, and test sets are created from the correspondingly sampled train, validation, and test sets of such split QMNIST.

In our experiments, we are evaluating the effect of different percentages of key instances in positive bags in the data on the success of automated detection and interpretation. We expect this to be related to the impact of the ratio of malignant cells (responsible for the diagnosis) and the total number of cells in a bag, on the correctly made diagnosis. When creating PAP-QMNIST experimental data, we can define the number of key instances in a bag (patient sample). Ideally, this number should mimic the number of key instances for a patient in the OC Dataset. However, it is virtually impossible to estimate the percentage of key instances in the OC Dataset. Therefore, we make experiments with different versions of PAP-QMNIST. First, we assume an ideal case with the same percentage of key instances for all patients, and we set the percentage of key instances for each bag to 5, 10, 20, or 30%. Second, to introduce variations that could be observed in real data, we design experiments with PAP-QMNIST where the percentage of key instances varies in positive bags and is sampled according to a beta distribution (commonly considered a suitable model for the random behavior of percentages and proportions) with the mean of 17.5% (the middle of the range [5 − 30%]) and standard deviations of 5% and 10% (Fig 3). These datasets are referred to as *PAP-QMNIST-beta1* and *PAP-QMNIST-beta2*, respectively. Parameters $\alpha$ and $\beta$ for beta distribution with a standard deviation of 5% are respectively equal to 9.932 and 46.82, and for beta distribution with a standard deviation of 10% are 2.352 and 11.086 respectively.

We choose the digit "4" to represent key instances in the PAP-QMNIST data; this provides a suitably difficult problem, where rotated versions of this digit may look very similar to several other digits in the data set, e.g., digits 7 or 2 depending on the writing style. The key instances are present *only* within positive bags, i.e., those representing samples from patients with malignancy.

## Experimental setup

The OC Dataset is intrinsically a MIL dataset, since it is divided into bags that correspond to patients. The SIL format of the OC Dataset is created by assigning weak bag-level positive labels to all images from positive bags while all images from negative bags are assigned weak bag-level negative labels. All the versions of PAP-QMNIST dataset, with different percentages of key instances in positive bags, are first generated as MIL datasets, i.e., images are sampled into bags; after that, the SIL format of the versions is created in the same way as the SIL format of OC Dataset. 24 bags of PAP-QMNIST are sampled once (for each chosen percentage of key instances in positive bags) and then, in the same way as bags of the OC Dataset, permuted for different folds.

We use 9-fold cross-validation for all methods and for both PAP-QMNIST and OC Dataset, where data is separated on the bag (patient) level. For each fold6 negative bags and 6 positive bags are used for training, 2 negative bags and 2 positive bags for validation, and 4 negative

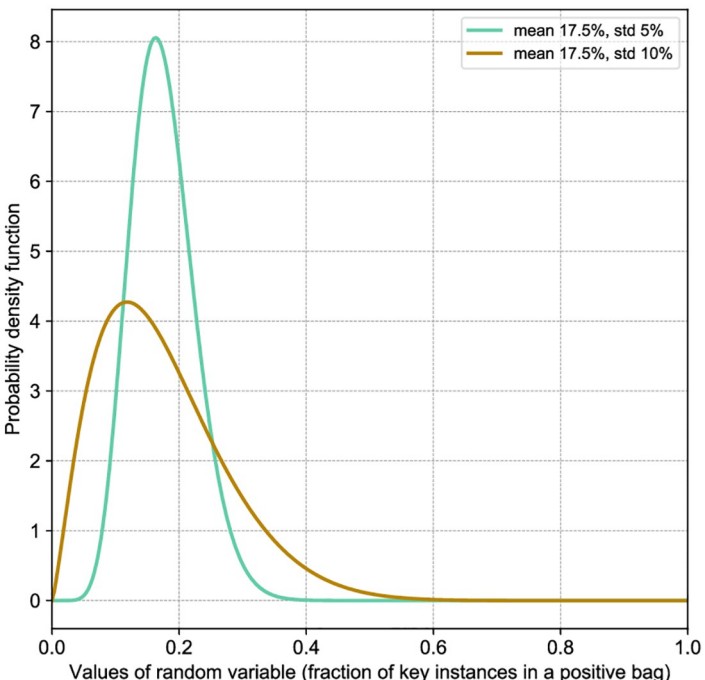

**Fig 3. Used beta distributions, defining the fraction of key instances in positive PAP-QMNIST bags.**

bags and 4 positive bags for testing. It always holds that in the test set, there are no instances from bags used in training or validation. Each bag appears in the test set of three folds with different combinations of bags for training, validation, and testing.

All experiments are performed using PyTorch [17]. The Albumentations [18] package is used to augment both PAP-QMNIST and OC Dataset during training and PAP-QMNIST during creation. Augmentations used during training are horizontal and vertical flips, rotation by multiples of 90 degrees (thereby avoiding interpolation), and added Gaussian noise. Images are standardized by the mean and standard deviation of the intensities in each of the three color channels over the whole dataset, for both PAP-QMNIST and OC Dataset. For both considered methods (ABMIL with sampling and SIL), and both datasets (PAP-QM-NIST and OC Dataset), we evaluate three different architectures: LeNet architecture as in Ilse et al. [9], ResNet18 [19] and SqueezeNet [20] (see the details of models in S1 Appendix). The choice of these architectures is made considering the size of images and their preliminary performance on PAP-QMNIST data. Optimization parameters for training SIL and MIL models are provided in S4 Table. Learning rates and weight decay regularization coefficients were selected for different architectures based on optimization on a subset of PAP-QMNIST.

During training, we store the model weights for each epoch. To finally select the model to use at inference, we observe the moving average (over a window of 15 epochs) of F1 score for SIL, and classification error for ABMIL with sampling, on the validation set. For the moving average window with the highest average F1 score or lowest classification error (for SIL and ABMIL with sampling, respectively), we select the model which provides the highest validation F1 score/the lowest classification error within that window. We save model weights no earlier than epoch 20 (for both SIL and ABMIL with sampling) to avoid possible fluctuations at the beginning of training.

For SIL, we set mini-batch size to 56 images and the maximum number of training epochs to 150 for both PAP-QMNIST and OC Dataset, which is defined beforehand, observing that the convergence to a maximum F1 score on the validation data is reached earlier than this number of epochs. A more detailed analysis on the effect of mini-batch sizes on the performance of SIL can be found in S2 Appendix.

For ABMIL with sampling, we perform experiments with mini-bag sizes with 2500, 1200, and 500 instances, corresponding to 26.9%, 12.9%, and 5.4% of the average number of instances per bag. We use a model parallel technique, sending different parts of the model to two different GPUs, each of 12, 16, or 32 GB of VRAM, depending on the mini-bag size and network architecture. The approximate number of test evaluations for each instance (see Koriakina et al. [14]) is calculated based on the bag with the highest number of instances, for which we make approximately 10 test evaluations per instance. We set the maximum number of training epochs to 1500, having observed beforehand that the best validation performance is reached well before this number.

**Quantitative evaluation.**    We perform quantitative evaluation of the observed algorithms by computing:

1a, accuracy at the bag ("patient") level for PAP-QMNIST;

1b, precision at the instance ("cell") level, *Precision@$K_i$* (a.k.a. crossover point of precision and recall), for the instances with the $K_i$ top attention weights/prediction scores, where $K_i$ is the number of key instances in a bag *i* for PAP-QMNIST;

2, accuracy at the patient (bag) level for OC Dataset.

In all cases, we observe different bag sizes for ABMIL with sampling and different percentages of key instances in positive bags in the bags. Accuracy is chosen as a standard metric for a balanced number of bags ("patients") in classes. The choice of *Precision@$K_i$* is driven by the possible application in clinical routine. A shortcoming of DL-based weakly supervised approaches is that they do not have a certain threshold of defining whether the instance is detected as positive or not, but rather a continuous range of values for weights/scores. In the absence of per-cell annotations, it would be infeasible for a cytotechnologist to examine all the instances detected as positive at several different thresholds. Instead, a limited number of instances with the top weights/scores would assist a cytotechnologist in decision making, provided that these top instances are diagnostically relevant for oral cancer. Therefore, in our study, we quantify the ability of DL-based approaches to detect the real key instances within top $K_i$ instances in a bag *i*.

For ABMIL with sampling, we calculate accuracy at the bag (patient) level in the same manner as in Koriakina et al. [14], by computing a majority voted mini-bag label for each test bag. Metric at the instance level is calculated only for the positive (containing images of the digit "4") bags in the test set of PAP-QMNIST, following [14]. Due to sampling, we aggregate an average attention weight for each instance. First, we store attention weights for an instance from all mini-bags where this instance is present and then compute its average attention weight using only attention weightsfrom mini-bags with the same predicted bag label as the majority bag label for all mini-bags where this instance is present. The attention weights are normalized for each mini-bag to the range [0, 1]. An instance in bag *i* is considered to be a correctly detected key instance when all of the following holds: (i) the predicted bag label is positive, (ii) the real instance label is positive, and (iii) the value for the average attention weight for the instance falls among top $K_i$ average attention weights of the bag *i*, where $K_i$ is the number of key instances in bag *i*.

To make the comparison between ABMIL with sampling and SIL as fair as possible, we compute performance metrics for SIL in the same way as we compute them for ABMIL: first, we obtain bag-level metric and then use bag labels together with instance labels to compute instance level metric. The threshold $t_f$ for each fold $f$, separating positive and negative test bags, is chosen as the middle value between the fifth percentile (calculated using linear interpolation) of the percentage of instances classified as positive (softmax score>0.5) in positive and negative bags of fold $f$ from train and validation sets. As an assisting technology to an expert, the DL-based methods should detect all questionable slides (bags) not to miss malignancy. A medical expert would then resolve challenging cases and provide a final decision. We choose the fifth percentile to reduce the risk of false negative bag predictions and, hence, to approach a relatively sensitive bag detection.

We then compute the SIL instance-level metric for the PAP-QMNIST test set by using prediction scores from the softmax layer. Due to the different nature of ABMIL with sampling and SIL, bags classified as negative by SIL may contain instances which are classified as positive, whereas bags classified as negative by ABMIL with sampling may not, by definition, contain any positive (i.e., key) instances [9]. For a fair methods comparison, we compute key instance detection performance of SIL in a similar way as for ABMIL with sampling, i.e., taking into account only the instances coming from bags with a real positive label. An instance in bag $i$ is considered to be a correctly detected key instance when all of the following holds: (i) the predicted bag label is positive, (ii) the real instance label is positive, and (iii) the softmax score for the instance falls in top $K_i$ scores of that bag $i$, where $K_i$ is the number of key instances in bag $i$. (Note that a key instance (softmax among top $K_i$ in a positively predicted bag) is not the same as an above-mentioned instance classified as positive (softmax score>0.5)).

**Qualitative evaluation.**   In addition to quantitative evaluation, we perform a visual inspection of top-ranked key instances detected in PAP-QMNIST and OC Datasets by both considered methods. We observe instances with the highest attention weights/prediction scores for each correctly detected positive bag (actually positive and also detected as positive (TP)) by the considered methods, trained and evaluated on the same data. We analyze the top 25 such instances detected in one of the folds of all generated versions of the PAP-QM-NIST dataset.

Qualitative evaluation of results on the OC Dataset is performed with the participation of an expert cytotechnologist. Our aim is to identify if there exists a correspondence between the key instances identified by the automated system (based on each of the models) and those identified by a human expert. For each patient in the test set, the cytotechnologist is presented with the 36 instances with the highest attention weights/prediction scores, as identified by the system in three folds of the OC data, and is asked toidentify images/cells with characteristics which can be related to malignancy. Cytological annotation is performed according to the Bethesda system [21, 22], where ASC-US stands for 'atypical squamous cells of unknown significance' and ASC-H for 'atypical squamous cells, cannot exclude a high-grade lesion'.

The images shown to our expert cytotechnologist are cut-outs of a larger size, $170 \times 170$ pixels, instead of the $80 \times 80$ pixels used by the methods, to make the examination more reliable (since such a setup is more similar to what cytotechnologists typically use). This difference does not affect the study, since we do not compare human performance to the performance of the considered methods, but we are interested in obtaining the opinion of an expert on the relevance of the cells the methods report as being most related to the malignant class.

## Results and discussion

### PAP-QMNIST dataset

We show accuracy at the bag level and *Precision@$K_i$* at the instance level for all considered methods in Fig 4 and present mean and standard deviation of metrics in Tables 1 and 2. For PAP-QMNIST with the fixed percentage of key instances, both metrics, averaged over three architectures, have higher values with 10%, 20% and 30% of key instances per positive bag, than with 5% of key instances per bag. We also observe that the training is more sensitive to hyper-parameter tuning at 5% key instances.

For ABMIL with sampling, we observe increased accuracy at the bag level and *Precision@$K_i$* when the percentage of key instances in positive bags increases. We see that high accuracy at the bag level corresponds to relatively high *Precision@$K_i$* for this method. The theory of MIL defines that one key instance in a bag makes the bag positive, however, we observe when training on PAP-QMNIST that one key instance is not enough for ABMIL with sampling to classify bags reliably, but rather some percentage (in our experiments, 10% or more) of key instances per positive bag is required. The accuracy at the bag level and *Precision@$K_i$* at the instance level for ABMIL, reach higher values for mini-bags of size 500 instances, on average, compared to ABMIL with larger mini-bags. This is in agreement with the previous study by Koriakina et al. [14].

Performances of both SIL and ABMIL with sampling on *PAP-QMNIST-beta1* are mainly similar to the performance of PAP-QMNIST with 20% of key instances, except the decreased performance of LeNet and ResNet18-based SIL. On *PAP-QMNIST-beta2*, each of the methods reach the same values for bag-level accuracy, but with the feature extractor based on different

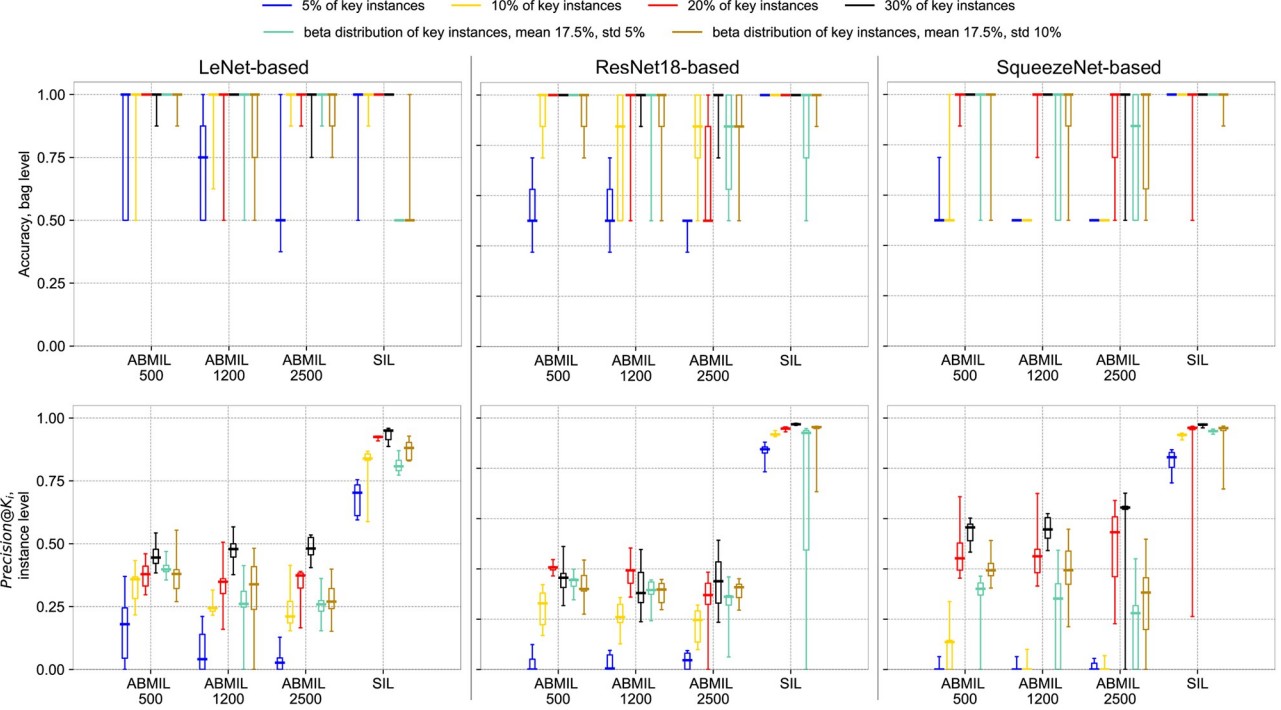

**Fig 4. Test set performance for ABMIL with sampling and SIL on PAP-QMNIST.** Top: Accuracy at the bag level. Bottom: Precision at the instance level, *Precision@$K_i$*, for instances with the $K_i$ top attention weights/prediction scores, where $K_i$ is the number of key instances in bag $i$. Three mini-bag sizes, 500, 1200, and 2500, used for ABMIL with sampling are indicated on the *x*-axis. Larger values of accuracy and *Precision@$K_i$* are better. The box plots display minimum, first quartile, median, third quartile, and maximum (the five-number summary) over 9 folds.

**Table 1. Accuracy at the bag level of the ABMIL with sampling and SIL methods on PAP-QMNIST test data.** Three mini-bag sizes, 500, 1200, and 2500, used for ABMIL with sampling are indicated in the column headings. The mean and standard deviation of the metric are computed over 9 folds. Bold face indicates the best performance for each dataset.

| Percentage of key instances in positive bags | | ABMIL 500 | ABMIL 1200 | ABMIL 2500 | SIL |
|---|---|---|---|---|---|
| 5% | LeNet | 0.833±0.236 | 0.722±0.211 | 0.556±0.168 | 0.889±0.208 |
| | ResNet18 | 0.528±0.115 | 0.542±0.102 | 0.486±0.039 | **1.000±0.000** |
| | SqueezeNet | 0.528±0.079 | 0.500±0.000 | 0.500±0.000 | **1.000±0.000** |
| 10% | LeNet | 0.944±0.157 | 0.958±0.118 | 0.972±0.052 | 0.986±0.039 |
| | ResNet18 | 0.944±0.086 | 0.806±0.221 | 0.819±0.168 | **1.000±0.000** |
| | SqueezeNet | 0.569±0.157 | 0.500±0.000 | 0.500±0.000 | **1.000±0.000** |
| 20% | LeNet | **1.000±0.000** | 0.944±0.157 | 0.986±0.039 | **1.000±0.000** |
| | ResNet18 | **1.000±0.000** | 0.889±0.208 | 0.694±0.221 | **1.000±0.000** |
| | SqueezeNet | 0.986±0.039 | 0.972±0.079 | 0.861±0.208 | 0.944±0.157 |
| 30% | LeNet | 0.986±0.039 | **1.000±0.000** | 0.972±0.079 | **1.000±0.000** |
| | ResNet18 | **1.000±0.000** | 0.986±0.039 | 0.958±0.083 | **1.000±0.000** |
| | SqueezeNet | **1.000±0.000** | **1.000±0.000** | 0.944±0.157 | **1.000±0.000** |
| *PAP-QMNIST-beta1,* | LeNet | **1.000±0.000** | 0.944±0.157 | 0.986±0.039 | 0.500±0.000 |
| beta-distributed | ResNet18 | **1.000±0.000** | 0.917±0.167 | 0.819±0.205 | 0.875±0.186 |
| mean = 17.5%, std = 5% | SqueezeNet | 0.944±0.157 | 0.792±0.236 | 0.764±0.239 | **1.000±0.000** |
| *PAP-QMNIST-beta2,* | LeNet | **0.986±0.039** | 0.875±0.186 | 0.931±0.086 | 0.569±0.157 |
| beta-distributed | ResNet18 | 0.931±0.086 | 0.903±0.153 | 0.847±0.193 | **0.986±0.039** |
| mean = 17.5%, std = 10% | SqueezeNet | 0.931±0.157 | 0.889±0.161 | 0.833±0.212 | **0.986±0.039** |

architecture types. SIL reaches higher values of the instance-level metric than ABMIL with sampling. The dissimilarity in bag accuracy for the different architecture types indicates that both methods may contribute to the analysis of similar data with a largely varying percentage of key instances in positive bags.

**Table 2. *Precision@$K_i$* at instance level of the ABMIL with sampling and SIL methods on PAP-QMNIST test data.** Three mini-bag sizes, 500, 1200, and 2500, used for ABMIL with sampling are indicated in the column headings. The mean and standard deviation of the metric are computed over 9 folds. Bold face indicates the best performance for each dataset.

| Percentage of key instances in positive bags | | ABMIL 500 | ABMIL 1200 | ABMIL 2500 | SIL |
|---|---|---|---|---|---|
| 5% | LeNet | 0.159±0.122 | 0.066±0.076 | 0.036±0.044 | 0.683±0.059 |
| | ResNet18 | 0.024±0.034 | 0.025±0.029 | 0.034±0.029 | **0.862±0.037** |
| | SqueezeNet | 0.009±0.017 | 0.010±0.019 | 0.012±0.017 | 0.827±0.047 |
| 10% | LeNet | 0.332±0.065 | 0.252±0.030 | 0.236±0.076 | 0.814±0.083 |
| | ResNet18 | 0.248±0.069 | 0.214±0.057 | 0.180±0.064 | **0.935±0.007** |
| | SqueezeNet | 0.084±0.088 | 0.017±0.031 | 0.011±0.020 | 0.931±0.008 |
| 20% | LeNet | 0.374±0.052 | 0.335±0.090 | 0.325±0.082 | 0.923±0.006 |
| | ResNet18 | 0.403±0.019 | 0.380±0.055 | 0.273±0.106 | **0.959±0.006** |
| | SqueezeNet | 0.469±0.095 | 0.455±0.107 | 0.499±0.150 | 0.876±0.235 |
| 30% | LeNet | 0.449±0.047 | 0.472±0.052 | 0.485±0.042 | 0.933±0.025 |
| | ResNet18 | 0.361±0.061 | 0.332±0.092 | 0.349±0.099 | **0.976±0.002** |
| | SqueezeNet | 0.544±0.047 | 0.558±0.048 | 0.567±0.207 | 0.972±0.004 |
| *PAP-QMNIST-beta1,* | LeNet | 0.401±0.030 | 0.261±0.106 | 0.256±0.058 | 0.814±0.029 |
| beta-distributed | ResNet18 | 0.348±0.036 | 0.305±0.053 | 0.260±0.085 | 0.711±0.354 |
| mean = 17.5%, std = 5% | SqueezeNet | 0.287±0.108 | 0.225±0.169 | 0.179±0.144 | **0.947±0.005** |
| *PAP-QMNIST-beta2,* | LeNet | 0.379±0.082 | 0.300±0.145 | 0.281±0.078 | 0.873±0.036 |
| beta-distributed | ResNet18 | 0.337±0.059 | 0.305±0.045 | 0.315±0.040 | **0.934±0.080** |
| mean = 17.5%, std = 10% | SqueezeNet | 0.403±0.062 | 0.387±0.126 | 0.286±0.150 | 0.931±0.076 |

Neither of the methods is able to classify all the bags of *PAP-QMNIST-beta2* correctly, however, both methods are able to classify all bags of *PAP-QMNIST-beta1*. This result could be related to the larger variability in the percentage of key instances in positive bags for *PAP-QMNIST-beta2*. The performance of LeNet-based SIL on PAP-QMNIST with beta-distributed percentage of key instances does not reach the performance of ResNet18 and SqueezeNet-based SIL, presumably due to the shallower architecture.

Somewhat unexpectedly, the accuracy values of the SIL approach are mostly higher or comparable to those of ABMIL with sampling, for the SIL and MIL each with the neural network architecture performing best on a PAP-QMNIST version. *Precision@$K_i$* values of the SIL approach surpass those of the MIL approach, for all the datasets and backbone architecture types. SIL based on ResNet18 and SqueezeNet reaches high values of accuracy and *Precision@$K_i$* also for datasets with (only) 5% of key instances.

Fig 5 illustrates the percentage of instances detected by SIL as positive for different architectures and different percentages of key instances per bag. The percent of instances classified as positive roughly corresponds to the real percent of key instances per bag for SqueezeNet and ResNet18-based SIL, matching the high *Precision@$K_i$* at the instance level in Fig 4 (bottom row). The percentage of instances detected as positive by LeNet-based SIL corresponds less to the real percent of key instances for versions of PAP-QMNIST dataset with fixed percentage of key instances and differs a lot from the real percent of key instances for *PAP-QMNIST-beta1* and *PAP-QMNIST-beta2*. This discrepancy is in accordance with the decreased bag-level performance of SIL based on LeNet as compared to SIL based on two other architectures.

Examples of instances with the highest attention weights/prediction scores, for PAP-QMNIST dataset with different percentages of key instances, from bags classified as positive by ABMIL with sampling and SIL, are presented in Fig 6. Test bags from PAP-QMNIST dataset with positive bag labels that are (incorrectly) classified as negative (ABMIL with sampling with mini-bag size of 500 instances on PAP-QMNIST with 5% of key instances) are not shown in Fig 6. One can observe that more diverse in color and rotation key instances are detected by SIL as compared to ABMIL with sampling, as well as that a higher number of actual key instances, among the instances with the highest attention weights/prediction scores, are found by SIL than by ABMIL with sampling. This observation is in agreement with the values of the instance metric *Precision@$K_i$* for SIL, which exceed the values of *Precision@$K_i$* for ABMIL with sampling for all the considered versions of PAP-QMNIST. The appearance of top key instances detected by ABMIL with sampling in the PAP-QMNIST dataset indicates that this method can be prone to paying attention to color and orientation, features which, for this dataset, are known (by design) to not be relevant for the classification task. Based on our results, we arrive at the conclusion that ABMIL with sampling has a tendency to focus on a subset of the present key instances (having the same shape, color, or orientation), whereas the SIL approach shows to be less prone to this type of mode collapse.

## Oral cancer dataset

Accuracy at the bag level for OC Dataset is shown in Fig 7, where accuracies of methods based on considered architectures, on average, are all above 0.75 (see Table 3). The accuracy of SIL is higher than the accuracy of ABMIL with sampling. The average accuracy of SIL exceeds 0.88 for all three architectures. The best-performing architecture, ResNet18, reaches a patient-level accuracy of 93.1%, which exceeds reported human experts' performance on this type of tasks and, therefore, shows great promise for future clinical use. The percentage of cells identified by SIL as coming from patients with OC for each of the 24 slides is illustrated in Fig 8.

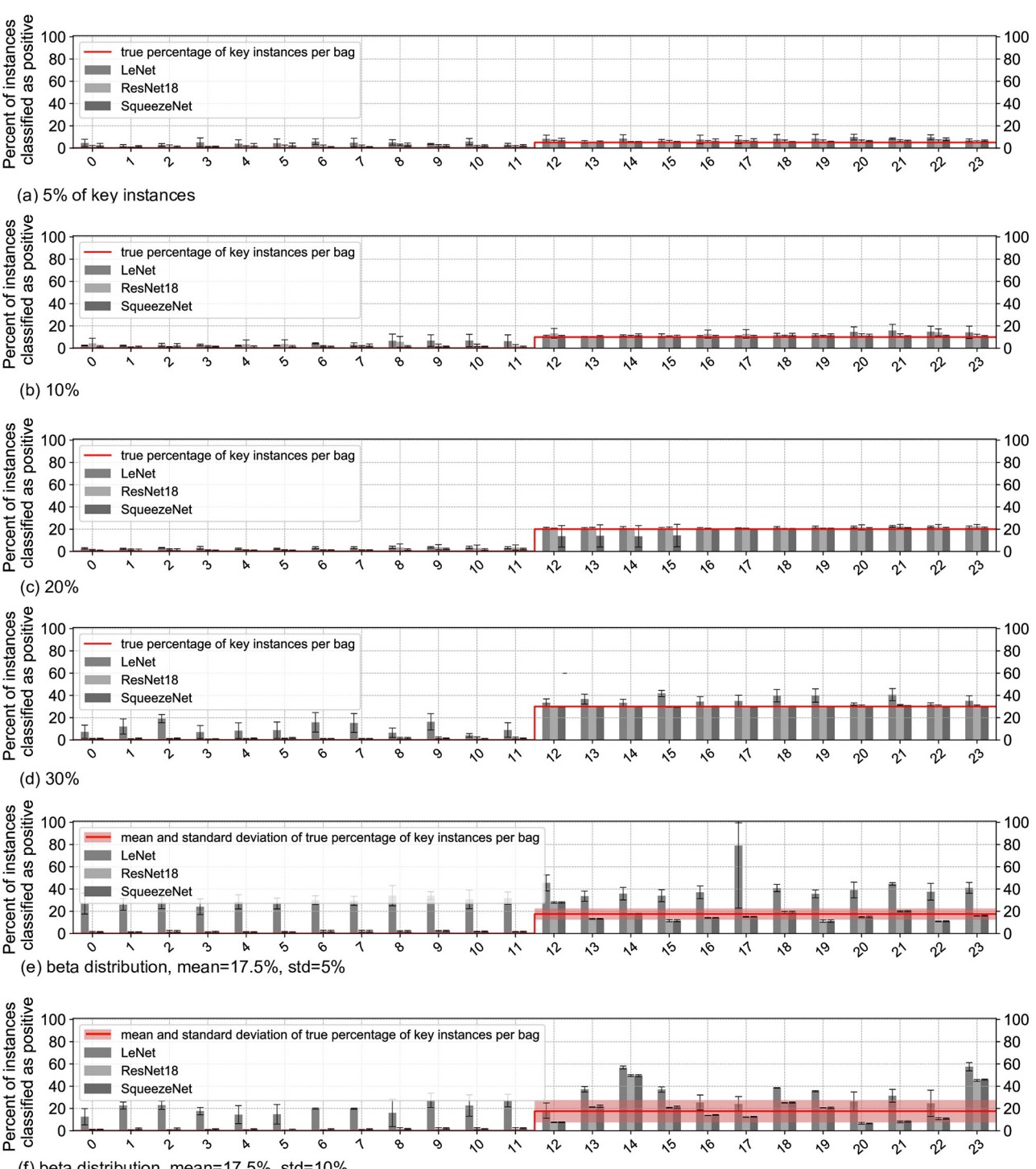

**Fig 5. Percentage of instances identified as positive by SIL approach on PAP-QMNIST dataset.** Instances identified as positive by the SIL approach have softmax score>0.5. Percentage is shown per each test bag for versions of the PAP-QMNIST dataset with different key instance ratios. Standard deviation is indicated by error bars and calculated for the 3 different folds in which each bag appears. Bags 0–11 are negative, and bags 12–23 are positive.

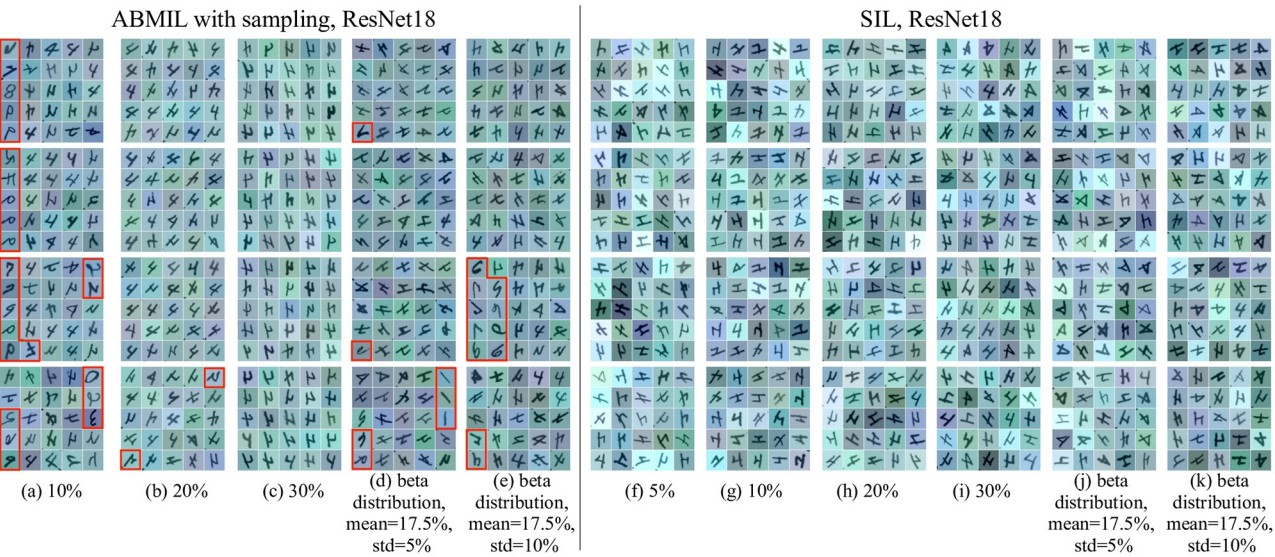

**Fig 6. Examples of instances detected by ABMIL with sampling and SIL approaches in the PAP-QMNIST dataset.** The top 25 instances with the highest attention weights/prediction scores for each of the four test bags with positive labels are shown, detected as positive in one of the folds of the PAP-QMNIST dataset. The percent of key instances per bag in the datasets is indicated under each subfigure. Red polygons delineate wrongly identified key instances—digits other than "4". (a)-(e) ABMIL with sampling based on ResNet18 (similar results observed for other architectures) and with minibag size of 500 instances. (f)-(k) SIL based on ResNet18.

Qualitative evaluation of the methods on OC Dataset at the instance level is performed involving an expert. Fig 9 shows four mosaics of 36 cells each, in a form presented to the cytotechnologist. We prepared 118 mosaics of cells from bags detected as positive (see Table 4). Each selection of 36 cells contains the instances (out of, on average, 9, 300 cells per patient) with the highest attention weights/prediction scores from the bags predicted as positive by the

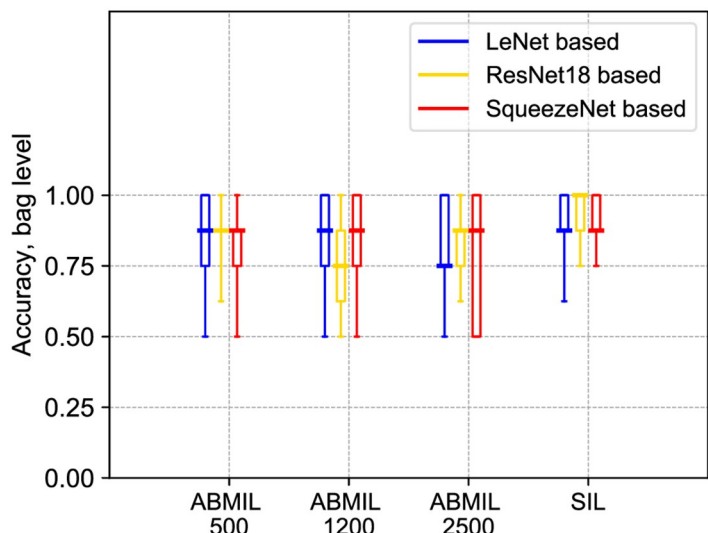

**Fig 7. Accuracy at the bag level of the ABMIL with sampling and SIL methods on the test set of OC Dataset.** Three mini-bag sizes, 500, 1200, and 2500, used for ABMIL with sampling are indicated on the *x*-axis. The box plots display minimum, first quartile, median, third quartile, and maximum (the five-number summary) over 9 folds.

**Table 3. Accuracy at the bag level of the ABMIL with sampling and SIL methods on the test set of OC Dataset.** The mean and standard deviation of the metric are computed over 9 folds. Bold face indicates the best performance.

|  | ABMIL 500 | ABMIL 1200 | ABMIL 2500 | SIL |
|---|---|---|---|---|
| LeNet | 0.847±0.174 | 0.847±0.163 | 0.806±0.178 | 0.889±0.116 |
| ResNet18 | 0.844±0.111 | 0.766±0.206 | 0.844±0.130 | **0.931±0.091** |
| SqueezeNet | 0.778±0.174 | 0.861±0.171 | 0.806±0.235 | 0.917±0.088 |

methods. The subsequent evaluation performed by the cytotechnologist resulted in 94 of these mosaics labeled as containing no suspicious cells but only cells from the categories: normal superficial cells (see Fig 9(a)), a mixture of normal superficial cells and normal intermediary cells, normal intermediary cells, or a mixture of normal superficial cells and cell debris or blood cells such as lymphocytes and neutrophils; 21 of the sets were annotated as benign but with reactive changes, (Fig 9(b)) or sets that contained a minority of not normal cells, such as

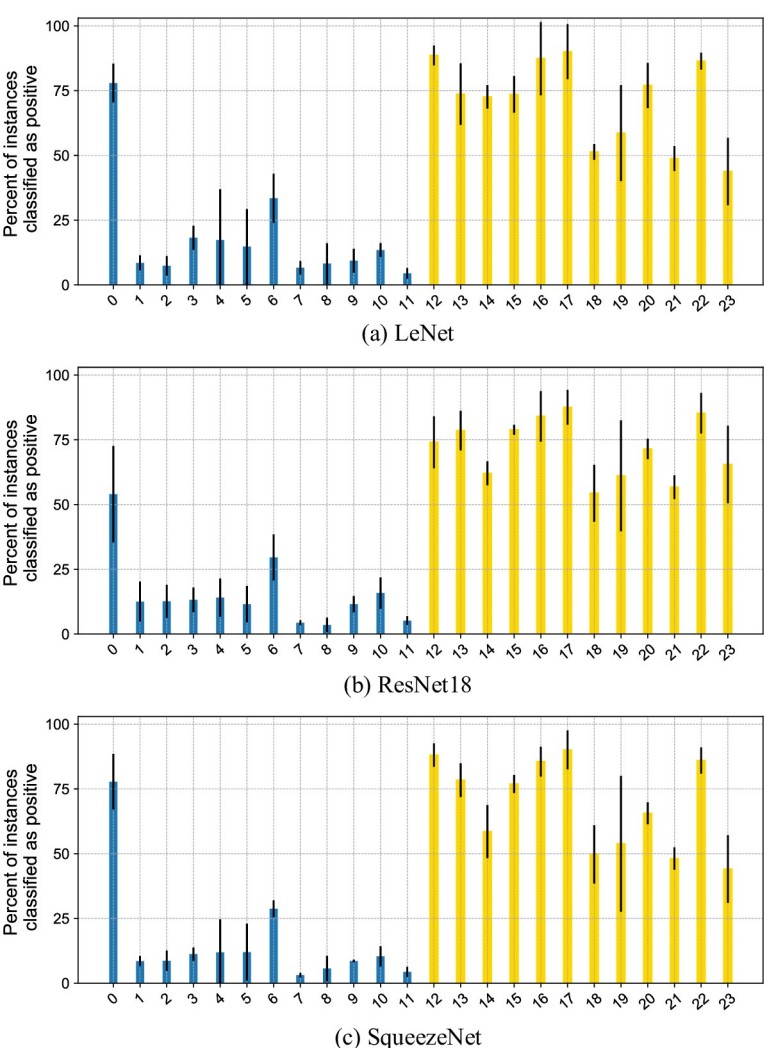

(a) LeNet

(b) ResNet18

(c) SqueezeNet

**Fig 8. Percentage of instances (cells) identified as positive by SIL approach on the OC Dataset.** Instances identified as positive by the SIL approach have softmax score>0.5. Percentage is shown per each test bag (patient). Error bars indicate the standard deviation over the 3 different folds in which each bag appears. Bags 0–11 (blue) are negative, and the rest (yellow) are positive.

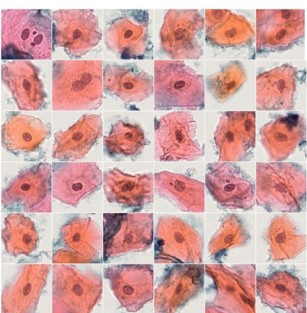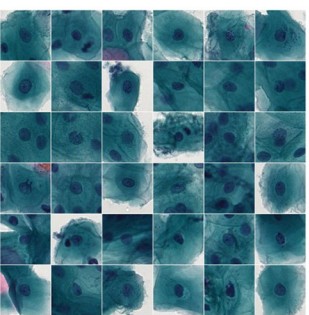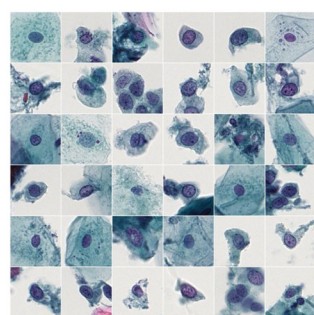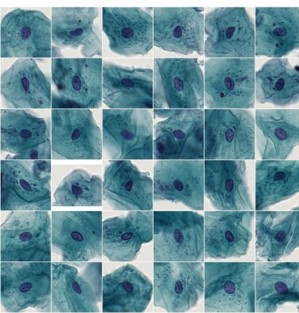

(a) Predicted as positive by ABMIL, all annotated as normal superficial cells by cytotechnologist

(b) Predicted as positive by ABMIL, several cells annotated as ASC-US by cytotechnologist

(c) Predicted as positive by SIL, several cells annotated as suspicious for high grade dysplasia by cytotechnologist

(d) Predicted as negative by SIL, annotated as normal intermediary cells by cytotechnologist

**Fig 9. Examples of instances detected by ABMIL with sampling and SIL approaches in the OC Dataset.** The top 36 instances with the highest attention weights/prediction scores on the test set of the OC Dataset are shown, together with the corresponding annotations by cytotechnologist.

metaplastic, ASC-US, or ASC-H, with remaining cells being normal. Three of the sets were annotated as suspicious for high grade dysplasia (see Fig 9(c)), all coming from one patient and detected by SIL with all three architectures. For reference, a set of cells predicted by the SIL approach as negative and annotated as normal cells are shown in Fig 9(d). For ABMIL with sampling, out of the mosaics with the top 36 detected key instances from bags predicted as positive, 21% contained cells annotated as reactive or not normal (for ABMIL with mini-bag size of 2500 instances—the best performing mini-bag size at the qualitative evaluation on OC Dataset), while out of the mosaics with key instances detected using SIL, 28% contained cells annotated as reactive or not normal.

The majority of sets of instances with the highest attention weights/prediction scores from the bags predicted as positive by the methods are annotated by the cytotechnologist as normal superficial cells or cells/debris not related to malignancy. PAP-stained superficial cells have a distinct orange/pink color and are not specific for malignancy but can be present both for healthy patients and for patients with OC. A possible explanation for why such cells are predicted as key instances is the fact that they do appear more frequently within patients with malignancy than within healthy patients for the in-house collected OC Dataset; in that sense, the models have correctly found a relation, however, the medical expertise finds this relation as not trustworthy for a decision. This highlights the importance of interpretable approaches that are the focus of our study. Among the sets of predicted key instances, there are also many cells which, although not indicating clear malignancy, do exhibit some deviation from normality. Such cells are detected by both ABMIL with sampling and SIL, and might be relevant for

**Table 4. Qualitative evaluation of ABMIL with sampling and SIL methods on OC Dataset.** Agreement of methods' prediction with judgments made by cytotechnologist on the top-ranked instances. See text for details.

|  | Predicted bag class for evaluated mosaics | | Mosaics containing suspicious cells according to cytotechnologist | |
|---|---|---|---|---|
|  | **negative** | **positive** | **fraction** | **percent** |
| ABMIL 500 | 11 | 25 | 3/25 | 12% |
| ABMIL 1200 | 7 | 29 | 5/29 | 17% |
| ABMIL 2500 | 8 | 28 | 6/28 | 21% |
| SIL | 0 | 36 | 10/36 | 28% |
| Total number | 26 | 118 | | |

early detection of OC, e.g., suggesting further medical investigation. The SIL approach succeeded to highly rank cells annotated as *high grade dysplasia*, indicating the feasibility of inferring cell-level information from patient-level annotations. These, for a diagnosis of highly relevant cells, were not ranked among the top 36 key instances using ABMIL with sampling for the same annotated data. Note that we do not consider the visual inspection of only 36 cells out of ca. 10,000 cells on a whole slide to correspond to any envisioned realistic assessment, but it is a means for us to compare the considered methods.

Considering that a human expert performs labeling of the cells using larger cut-outs than those used by the automated approaches, one may argue that the evaluated methods could also be applied to larger cut-outs of the original image. Classification of larger crops is computationally more demanding, especially for the ABMIL with sampling method, and, more importantly, larger crops contain bigger parts of neighboring cells, which shifts the focus from a particular cell of interest. The cytotechnologist can distinguish clutter from cells and explicitly focus on the central one, whereas computer methods may take additional clutter as a feature, which leads to reduced interpretability of the results. We do not see any reason to believe that a change to larger cut-outs would change the relative performance of the observed methods.

**In summary,** our results indicate that SIL outperforms ABMIL with sampling for both synthetic and real world data. This stands in contrast to a common assumption that MIL is more suitable than SIL for learning from weakly annotated data. Possibly, our results are due to the nature of the here addressed task, with rather few and very large bags of varying sizes. The consistency of results for both real and synthetic datasets suggests that PAP-QMNIST can be useful during preliminary studies of method behavior before the analysis of real world cytology data is initiated.

## Summary and conclusion

In this study, we investigate two different weakly supervised DL methods with the aim to reach reliable and interpretable OC detection: ABMIL with within-bags sampling, an approach that belongs to the MIL family of methods, and a more conventional SIL classification. We evaluate performance on PAP-QMNIST—a synthetic dataset mimicking, in terms of several parameters relevant to our analysis, in-house OC cytological data, as well as on real OC data from 24 patients. On the synthetic PAP-QMNIST data, we explore how the percentage and distribution of key instances in positive bags affect the classification performance.

An overall conclusion of the performed comparative study is that, for this task, we do not observe any reasons to use ABMIL with sampling instead of the less complex and less memory-demanding SIL approach. The exception would be datasets with largely varying percentages of key instances in positive bags, where ABMIL with sampling reaches similar performance as SIL on bag-level classification, however, with feature extractors based on different architecture types. This dissimilarity suggests that cytotechnologists could gain from both methods on such data. Nevertheless, SIL's ability to detect key instances surpasses, even on such datasets, the ability of ABMIL with sampling.

Through the performed experiments, we demonstrate that:

- It is possible not only to separate healthy patients from patients with malignancy, but also to detect cells related to malignancy, by utilizing only patient-level annotations on a fairly small number of patients;

- SIL, on average, outperforms a here observed representative of MIL methods—ABMIL with sampling. Obtained results are consistent on the two datasets, the real cytology data (OC Dataset), and the synthetic PAP-QMNIST dataset;

- The PAP-QMNIST dataset facilitated evaluation of instance-level performance, as well as analysis of interpretability of the methods, and allowed to observe a tendency of ABMIL with sampling to focus on, for the task, not relevant features.

The performed study is a valuable step towards the development of interpretable DL-based OC detection methods. Furthermore, we see a potential of the created synthetic PAP-QMNIST dataset for usage in other similar evaluation scenarios.

## Limitations and future directions

Artificial intelligence (AI) systems can potentially reduce error rates and promote knowledge discovery in healthcare and medicine without the need for accurate annotations. A common opinion in the AI community is that AI-based methods should be considered as tools that facilitate diagnostics by medical professionals but not as a replacement for humans, [23, 24] An integration into clinical practice of AI-based systems that would assist human experts still remains difficult partly due to the black-box nature of DL-based models. While in this study we evaluate DL-based methods for cytology that offer interpretability at the instance level, there is still a lag in achieving a full understanding of DL models behavior. The task of determining specific features in the image, e.g., color, shape, and texture, that are important to DL models for class prediction remains unsolved in the AI community.

A second limitation of our study is the limited size and lack of diversity in the OC dataset. Although the analysis of limited cytology data can be useful for similar data-scarce scenarios, the in-house OC dataset is collected from patients in one country and digitized by one scanner, which could negatively affect the ability of DL methods to generalize to data from new sites. This appears to be a general problem in the research community: even large benchmark datasets in healthcare, such as the Cancer Metastases in Lymph Nodes Challenge (CAMELYON) [25, 26] and the Cancer Genome Atlas (TCGA) [27] datasets, lack diverse representations, [28]

A remaining challenge in the community is to develop AI-based systems for fine-grained cancer type classification of cytology data and of oral cancer data in particular. In our study, we focus on detecting the presence of abnormality but not on defining oral cancer subclasses, leaving final precise diagnoses to experts. We remind that the task of cell subtype classification in cytology is subjective and laborious, which leaves room for further research.

We also would like to mention that this study is tailored for classification of cytology data, where there is no spatial relation among cells, and we make a choice of specific MIL and SIL methods accordingly. The classification of spatially related data, e.g., histology data, may benefit more from methods which consider such relation.

To summarize, there is a potential for applying AI-based systems for oral cancer detection, as shown in this study; however, there is still a long path until such systems are integrated into clinical practice for the detection of oral and other cancers.

## Supporting information

**S1 Dataset. PAP-QMNIST dataset.**
(PDF)

**S1 Table. Transformations applied to QMNIST for PAP-QMNIST dataset generation.**
(PDF)

**S2 Table. Architecture of LeNet-based ABMIL with sampling model (from [9]).**
(PDF)

**S3 Table. Architecture of LeNet-based SIL model.**
(PDF)

**S4 Table. The used optimization parameters for ABMIL with sampling and SIL for different neural network architectures.**
(PDF)

**S1 Appendix. Details of models for SIL and ABMIL with sampling.**
(PDF)

**S2 Appendix. The effect of mini-batch size on the performance of SIL observed on PAP-QMNIST.**
(PDF)

**S1 Fig. The performance of SIL trained for 1500 epochs on the test set of the PAP-QM-NIST.** Four mini-batch sizes are indicated on the x-axis. The box plots display minimum, first quartile, median, third quartile, and maximum (the five-number summary) over 9 folds.
(TIF)

## Acknowledgments

We are grateful to M.D. Christina Runow Stark for providing oral cancer samples and to M.D. Jan-Michaél Hirsch for initiating and continuously supporting the project on early detection of oral cancer. A part of the experiments was enabled by computational resources provided by the National Academic Infrastructure for Supercomputing in Sweden (NAISS) and the Swedish National Infrastructure for Computing (SNIC) at Chalmers Centre for Computational Science and Engineering (C3SE).

## Author Contributions

**Conceptualization:** Nadezhda Koriakina, Nataša Sladoje, Joakim Lindblad.

**Data curation:** Nadezhda Koriakina, Joakim Lindblad.

**Formal analysis:** Nataša Sladoje.

**Funding acquisition:** Joakim Lindblad.

**Investigation:** Vladimir Bašić.

**Methodology:** Nadezhda Koriakina, Nataša Sladoje, Joakim Lindblad.

**Project administration:** Nataša Sladoje, Joakim Lindblad.

**Resources:** Joakim Lindblad.

**Software:** Nadezhda Koriakina.

**Supervision:** Nataša Sladoje, Vladimir Bašić, Joakim Lindblad.

**Validation:** Nadezhda Koriakina, Nataša Sladoje, Vladimir Bašić.

**Visualization:** Nadezhda Koriakina.

**Writing – original draft:** Nadezhda Koriakina.

**Writing – review & editing:** Nadezhda Koriakina, Nataša Sladoje, Vladimir Bašić, Joakim Lindblad.

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
