## [Decision Letter · Decision Letter 0]

14 Sep 2023

PONE-D-23-20689Deep multiple instance learning versus conventional deep single instance learning for interpretable oral cancer detectionPLOS ONE

Dear Dr. Koriakina,

Thank you for submitting your manuscript to PLOS ONE. After careful consideration, we feel that it has merit but does not fully meet PLOS ONE’s publication criteria as it currently stands. Therefore, we invite you to submit a revised version of the manuscript that addresses the points raised during the review process.

We look forward to receiving your revised manuscript.

Kind regards,

Kathiravan Srinivasan

Academic Editor

PLOS ONE

Journal Requirements:

"This work is supported by: Sweden’s Innovation Agency (VINNOVA) https://www.vinnova.se/en/apply-for-funding/funded-projects/, grants 2017-02447 (J.L.) and 2020-03611 (J.L.), and the Swedish Research Council https://www.vr.se/english/swecris.html#/, grant 2017-04385(J.L.). A part of the computations was enabled by resources provided by the National Academic Infrastructure for Supercomputing in Sweden (NAISS) and the Swedish National Infrastructure for Computing (SNIC) at Chalmers Centre for Computational Science and Engineering (C3SE), partially funded by the Swedish Research Council through grant agreements no. 2022-06725 and no. 2018-05973. The funders had no role in study design, data collection and analysis, decision to publish, or preparation of the manuscript."

Please provide an amended statement that declares *all* the funding or sources of support (whether external or internal to your organization) received during this study, as detailed online in our guide for authors at http://journals.plos.org/plosone/s/submit-now.  

Please also include the statement “There was no additional external funding received for this study.” in your updated Funding Statement. 

"This work is supported by: Sweden’s Innovation Agency (VINNOVA), grants 2017-02447 and 2020-03611, and the Swedish Research Council, grant 2017-04385. A part of the computations was enabled by resources provided by the National Academic Infrastructure for Supercomputing in Sweden (NAISS) and the Swedish National Infrastructure for Computing (SNIC) at Chalmers Centre for Computational Science and Engineering (C3SE), partially funded by the Swedish Research Council through grant agreements no. 2022-06725 and no. 2018-05973."

Funding information should not appear in the Acknowledgments section or other areas of your manuscript. We will only publish funding information present in the Funding Statement section of the online submission form. 

"This work is supported by: Sweden’s Innovation Agency (VINNOVA) https://www.vinnova.se/en/apply-for-funding/funded-projects/, grants 2017-02447 (J.L.) and 2020-03611 (J.L.), and the Swedish Research Council https://www.vr.se/english/swecris.html#/, grant 2017-04385(J.L.). A part of the computations was enabled by resources provided by the National Academic Infrastructure for Supercomputing in Sweden (NAISS) and the Swedish National Infrastructure for Computing (SNIC) at Chalmers Centre for Computational Science and Engineering (C3SE), partially funded by the Swedish Research Council through grant agreements no. 2022-06725 and no. 2018-05973. The funders had no role in study design, data collection and analysis, decision to publish, or preparation of the manuscript."

5. We noted in your submission details that a portion of your manuscript may have been presented or published elsewhere:

"Preprint of old version in arxiv. Koriakina N, Sladoje N, Bašić V, Lindblad J. Oral cancer detection and interpretation: Deep multiple instance learning versus conventional deep single instance learning. arXiv preprint arXiv:2202.01783. 2022 Feb 3."

Please clarify whether this [publication] was peer-reviewed and formally published. If this work was previously peer-reviewed and published, in the cover letter please provide the reason that this work does not constitute dual publication and should be included in the current manuscript.

7. We note that you have stated that you will provide repository information for your data at acceptance. Should your manuscript be accepted for publication, we will hold it until you provide the relevant accession numbers or DOIs necessary to access your data. If you wish to make changes to your Data Availability statement, please describe these changes in your cover letter and we will update your Data Availability statement to reflect the information you provide.

8. Your ethics statement should only appear in the Methods section of your manuscript. If your ethics statement is written in any section besides the Methods, please move it to the Methods section and delete it from any other section. Please ensure that your ethics statement is included in your manuscript, as the ethics statement entered into the online submission form will not be published alongside your manuscript. 

**Additional Editor Comments:**

Please revise and resubmit your manuscript.  

Reviewers' comments:

Reviewer's Responses to Questions

**Comments to the Author**

1. Is the manuscript technically sound, and do the data support the conclusions?

Reviewer #1: Yes

Reviewer #2: Partly

Reviewer #3: Partly

Reviewer #4: Yes

Reviewer #5: Yes

2. Has the statistical analysis been performed appropriately and rigorously? 

Reviewer #1: Yes

Reviewer #2: Yes

Reviewer #3: No

Reviewer #4: Yes

Reviewer #5: No

3. Have the authors made all data underlying the findings in their manuscript fully available?

Reviewer #1: Yes

Reviewer #2: Yes

Reviewer #3: Yes

Reviewer #4: No

Reviewer #5: No

4. Is the manuscript presented in an intelligible fashion and written in standard English?

Reviewer #1: Yes

Reviewer #2: Yes

Reviewer #3: Yes

Reviewer #4: Yes

Reviewer #5: No

5. Review Comments to the Author

Reviewer #1: The authors created a new database to use for detection of malign cells correlated with oral cancer (OC). Since annotation on the cell level seems near impossible, the authors augmented QMNIST to emulate the characteristics of malign cells.

Comments:

1) In the abstract, you mention "This dataset serves as a proxy...". This needs to be better explained and describe what you aim to achieve and how you use it.

2) Have the authors considered self-supervised methods? Usually there are improvements when combined with MIL approaches. Some results even with simple methods would benefit the paper.

3) line 58 "comprise" should be "include"

4) Phrase in lines 64-68 should be rewritten to convery the meaning in a more clear manner.

5) In lines 82-83 you write: "A main advantage of PAP-QMNIST is that is offers access to reliable GT annotation at the instance (cell) level". In the previous sentence, you write that GT annotation is not feasible. How both are true? Giving a brief overview of the dataset you created and what you aim to achieve would be beneficial here. Moreover, being more "precise" in the language would avoid confusion (it seems that PAP-QMNIST does not have cells at all).

6) In the end of Introduction, I believe "splitting" the contributions and findings, as well as adding a small overview of your findings would be beneficial for the reader (to have a first impression on what to expect from the rest of the paper).

7) Is there a reason for using F1 score instead of metrics like FAR, FRR? These metrics are interpreted in a manner that is easier to associate a cost for each action. An input of a medical expert, e.g. we are interested in very low number of False Negatives, would also add value to your findings.

8) In section "Considerd MIL method" some description/refresher would help the flow of the paper. For example, what is the modified version described in [13] (again a short description and the reader can refer to [13] for more details)? But throught the paper there isn't any model description.

9) In line 177 you mention 24 patients, and in lines 185-186 you give some extra information. It would help the reader

if you transferred the info in 185-186 in line 177.

10) In line 194 you write: "Reliable cell-level annotations ... are scarce". In line 199 you write: "while having reliable GT ... at the instance (cell) level. Same as comment 5). This creates confusion.

11) In line 204 you cite some other datasets. Since your main contribution is the creation of PAP-QMNIST, it would enhance your case if you would show comparative performance of a classification method(s)based on your dataset and those you cited.

12) In line 215 you write "transformations" expected in OC. It would the reader if you described those, or if you cited a paper that does.

13) Have you tried any AI colorization scheme? There are some free tools for this as well. Would be interesting to see if they have any effect compared to your way of colorization the QMNIST images.

14) In lines 221-222 you write: "...give colorization visually far from colors of OC data". How do you check this? Please include it in paper.

15) In line 225 you write: "if color appears as an important feature... over-fitted the data". Couldn't be that color is an important feature in general and has predictive power? Have you compared the performance with greyscale images? The way you present it is that if color ends up being important then there is over-fitting, which makes one wonder why bother with it then?

16) You need to explain your claim in 236-239. The impact of the ratio could differ between different classification schemes.

17) Based on your claim in lines 254-258 wouldn't the digit 6 or 9 make a better candidate than 4?

18) In line 262 "positive labels" refer to images with digit 4 (though all the tranformations)?

19) In line 267 you write:"For each key instance setting". What are the different "settings"?

20) In line 21 you mention F1 score again. Please consider including other metrics like TPR, FRR, etc.

21) In line 295 you mention that you trained the model for 20 epochs. At first glance, this seems like a low number. Have you tried training for 50 or 100 epochs to see if the behavior changes?

22) In line 315 you mention crossover of precision and recall. Since this happens at the instance level wouldn't your data be skewed towards negative samples making precision unreliable? Why did you choose that?

23) In line 321, is sampling with replacement or without?

24) In lines 330-334, it is not clear why you impose extra conditions. An instance in bag i is considered to be correctly detected as key instance when i and ii and iii hold true. But ii writes that the true label is positive. Isn't that information alone sufficient??

25) In line 371 put the phrase "for each patient in the test set" at the beginning of the sentence.

26) In line 381, you write you do not compare human performance .... Why not? Wouldn't that be useful? Such a method could be used as a substitute in places with lack of experts.

27) The results section should be combined with the Discussion section to highlight its findings. Currently, it seems that it "aggregates" the captions of tables and figures without offering any insights on what they mean and what value they add to the story of the paper.

This paper would benefit if the authors made a more "concetrated" effort to highlight their contribution, which is the dataset they created and how it can be used to train systems able to detect OC. If they reduced/reorganized figures and tables, it would enable focusing on the dataset and networks they used, which in turn would highlight their findings.

Reviewer #2: Dear authors,

I read with great interest the manuscript, which falls within the aim of this Journal. In my honest opinion, the topic is interesting enough to attract the readers’ attention. Nevertheless, authors should clarify some points and improve the discussion, as suggested below. Authors should consider the following recommendations:

In my opinion you have to refer in the paper to the updated literature about how the technique can be usefull for oral cancer detection as well for ohter cancer endometrial cancer and in general for all malignanices .

Also in infertility as has been demonstrated how machine learning can be helpfull.

I suggest you to read and cite these articles:

Circulating miRNAs as a Tool for Early Diagnosis of Endometrial Cancer-Implications for the Fertility-Sparing Process: Clinical, Biological, and Legal Aspects

Fertility-Sparing Strategies for Early-Stage Endometrial Cancer: Stepping towards Precision Medicine Based on the Molecular Fingerprint

The Future Is Coming: Artificial Intelligence in the Treatment of Infertility Could Improve Assisted Reproduction Outcomes-The Value of Regulatory Frameworks

Endometrial Cancer in Reproductive Age: Fertility-Sparing Approach and Reproductive Outcomes

Reviewer #3: This study seeks to explore AI-based methods for oral cancer (OC) detection, aiming to provide a less invasive alternative to histological examination. By comparing conventional single instance learning (SIL) and modern multiple instance learning (MIL) approaches using real OC data and a synthetic dataset (PAP-QMNIST), the study attempts to determine their effectiveness in identifying malignant and dysplastic cells. However, the study's scope is limited to performance evaluation and lacks a comprehensive analysis of the potential challenges and practical implications of implementing AI-driven OC detection in clinical settings. Additionally, while it compares SIL and MIL methods, it does not explore other AI-based approaches or address the need for interpretability and validation in real clinical scenarios, leaving room for further research in these critical areas.

Reviewer #4: 1) Avoid the extensive use of words like "we", "our"

2) Don't mention the URL of the code in abstract section

3) The introduction section is too long. It is recommended to summarize this section.

4) Figures and Tables captions are too long (in most cases)

5) The number of patients (24) is limited for a comparative study. Explain how could you overcome this limitation.

6) Why have you chosen certain measures of performance and neglected other measures? Justify

7) Give a comparison (in tabulated form) with previous studies in the field (before conclusion section)\\

8) Add (3-5) recent references 2022-2023

Reviewer #5: Please find the detailed comments as follows:

1. Authors have conducted experiments on a very limited and outdated set of deep learning models, which is insufficient in the current scenario.

2. Merely mentioning ResNet is not enough; it should clearly state which version of the ResNet model was used.

3. The manuscript lacks a state-of-the-art comparison with other models, such as DenseNet 201, EfficientNet-B0 to B7, MobileNet v3, etc.

4. There are very limited details about the deep learning models in the manuscript. Please provide more details.

5. Architectural diagrams of the deep learning model are also missing.

6. The details of the hyperparameters in the manuscript are not sufficient to support the reproducibility of the results. Just providing the GitHub link is not enough; crucial details should be in the main manuscript.

7. The manuscript does not properly state the research gaps that motivated the proposed work.

8. There are various grammatical and typographical mistakes in the manuscript. Please correct them.

9. A well-designed block diagram to illustrate the complete methodology would enhance the understanding of the manuscript.

10. The literature review is quite outdated and not very significant. Please include recent papers. Authors are also encouraged to discuss how the proposed work significantly contributes in the context of those recent works.

11. Authors should state the limitations of the proposed work and provide directions for future research.

6. PLOS authors have the option to publish the peer review history of their article (what does this mean?). If published, this will include your full peer review and any attached files.

Reviewer #1: No

Reviewer #2: **Yes: **Giuseppe Gullo

Reviewer #3: No

Reviewer #4: **Yes: **Hossam El-Din Moustafa

Reviewer #5: **Yes: **Rakesh Chandra Joshi

---

## [Author Response · Author response to Decision Letter 0]

5 Dec 2023

We thank the reviewers and academic editor for carefully reviewing our manuscript. 

Journal Requirements:

Authors’ response: We (1) removed excessive information of the current address, (2) changed the order of the section Acknowledgements and section Author summary, (3) changed S1 Fig. format in the Supporting information section, (4) changed the format of writing from ‘Fig. 1’ to ‘Fig 1’, (5) renamed ‘S1_fig.tiff’ to ‘S1_Fig.tif’, (6) changed from ‘Dept.’ to ‘Department’, (7) changed section title from ‘Author summary’ to ‘Author Contributions’, (8) added initials ‘(NK)’ of the corresponding author after email, (9) use math symbol instead of italics for Recall@Ki .

Authors’ response: We updated the Funding Information response to remove sources that are not sources of funding for the work included in this submission. The computational resources provided by NAISS/SNIC are funded separately from the submitted work. The updated Funding Statement is: “This work is supported by: Sweden’s Innovation Agency (VINNOVA) https://www.vinnova.se/en/apply-for-funding/funded-projects/, grants 2017-02447 (J.L.), 2021-01420 (J.L.), and 2020-03611 (J.L.), and the Swedish Research Council https://www.vr.se/english/swecris.html#/, grant 2017-04385 (J.L.). The funders had no role in study design, data collection and analysis, decision to publish, or preparation of the manuscript. There was no additional external funding received for this study.”

"This work is supported by: Sweden’s Innovation Agency (VINNOVA) https://www.vinnova.se/en/apply-for-funding/funded-projects/, grants 2017-02447 (J.L.) and 2020-03611 (J.L.), and the Swedish Research Council https://www.vr.se/english/swecris.html#/, grant 2017-04385(J.L.). A part of the computations was enabled by resources provided by the National Academic Infrastructure for Supercomputing in Sweden (NAISS) and the Swedish National Infrastructure for Computing (SNIC) at Chalmers Centre for Computational Science and Engineering (C3SE), partially funded by the Swedish Research Council through grant agreements no. 2022-06725 and no. 2018-05973. The funders had no role in study design, data collection and analysis, decision to publish, or preparation of the manuscript."

Please provide an amended statement that declares *all* the funding or sources of support (whether external or internal to your organization) received during this study, as detailed online in our guide for authors at http://journals.plos.org/plosone/s/submit-now. 

Please also include the statement “There was no additional external funding received for this study.” in your updated Funding Statement. 

Authors’ response: The computational resources provided by NAISS/SNIC are funded separately from the submitted work. The updated Funding Statement is: “This work is supported by: Sweden’s Innovation Agency (VINNOVA) https://www.vinnova.se/en/apply-for-funding/funded-projects/, grants 2017-02447, (J.L.), 2021-01420 (J.L.), and 2020-03611 (J.L.), and the Swedish Research Council https://www.vr.se/english/swecris.html#/, grant 2017-04385 (J.L.). The funders had no role in study design, data collection and analysis, decision to publish, or preparation of the manuscript. There was no additional external funding received for this study.”

"This work is supported by: Sweden’s Innovation Agency (VINNOVA), grants 2017-02447 and 2020-03611, and the Swedish Research Council, grant 2017-04385. A part of the computations was enabled by resources provided by the National Academic Infrastructure for Supercomputing in Sweden (NAISS) and the Swedish National Infrastructure for Computing (SNIC) at Chalmers Centre for Computational Science and Engineering (C3SE), partially funded by the Swedish Research Council through grant agreements no. 2022-06725 and no. 2018-05973."

Funding information should not appear in the Acknowledgments section or other areas of your manuscript. We will only publish funding information present in the Funding Statement section of the online submission form. 

"This work is supported by: Sweden’s Innovation Agency (VINNOVA) https://www.vinnova.se/en/apply-for-funding/funded-projects/, grants 2017-02447 (J.L.) and 2020-03611 (J.L.), and the Swedish Research Council https://www.vr.se/english/swecris.html#/, grant 2017-04385 (J.L.). A part of the computations was enabled by resources provided by the National Academic Infrastructure for Supercomputing in Sweden (NAISS) and the Swedish National Infrastructure for Computing (SNIC) at Chalmers Centre for Computational Science and Engineering (C3SE), partially funded by the Swedish Research Council through grant agreements no. 2022-06725 and no. 2018-05973. The funders had no role in study design, data collection and analysis, decision to publish, or preparation of the manuscript."

Authors’ response: We removed any funding-related text from the manuscript. Our updated Funding Statement is: “This work is supported by: Sweden’s Innovation Agency (VINNOVA) https://www.vinnova.se/en/apply-for-funding/funded-projects/, grants 2017-02447 (J.L.), 2021-01420 (J.L.), and 2020-03611 (J.L.), and the Swedish Research Council https://www.vr.se/english/swecris.html#/, grant 2017-04385 (J.L.). The funders had no role in study design, data collection and analysis, decision to publish, or preparation of the manuscript. There was no additional external funding received for this study.” 

5. We noted in your submission details that a portion of your manuscript may have been presented or published elsewhere:

"Preprint of old version in arxiv. Koriakina N, Sladoje N, Bašić V, Lindblad J. Oral cancer detection and interpretation: Deep multiple instance learning versus conventional deep single instance learning. arXiv preprint arXiv:2202.01783. 2022 Feb 3."

Please clarify whether this [publication] was peer-reviewed and formally published. If this work was previously peer-reviewed and published, in the cover letter please provide the reason that this work does not constitute dual publication and should be included in the current manuscript.

Authors’ response: The work [1] is an unreviewed preprint. It has not passed a peer review, and has not been formally published. We have introduced modifications into [1] and performed additional experiments before submitting the manuscript to PLOS ONE.

[1] “Koriakina N, Sladoje N, Bašić V, Lindblad J. Oral cancer detection and interpretation: Deep multiple instance learning versus conventional deep single instance learning. arXiv preprint arXiv:2202.01783. 2022 Feb 3.”

Authors’ response: In our study, we perform comparative evaluation of approaches on two datasets, synthetic PAP-QMNIST data and real world oral cancer data. The codes for generating synthetic PAP-QMNIST data are publicly available as well as the exact versions used in the submitted manuscript. The synthetic PAP-QMNIST data is essential to our study and the majority of the quantitative results and study findings reported in the article are obtained using PAP-QMNIST data. The oral cancer data are not publically available due to ethical permit restrictions. However, we will provide the values used to build graphs, the values behind the means, standard deviations, and other measures reported, as well as the weights for neural network models trained on oral cancer data. 

7. We note that you have stated that you will provide repository information for your data at acceptance. Should your manuscript be accepted for publication, we will hold it until you provide the relevant accession numbers or DOIs necessary to access your data. If you wish to make changes to your Data Availability statement, please describe these changes in your cover letter and we will update your Data Availability statement to reflect the information you provide.

Authors’ response: We will provide the values used to build graphs, the values behind the means, standard deviations, and other measures, as well as the weights for neural network models trained on oral cancer data and PAP-QMNIST data in Zenodo repository upon acceptance. Our preliminary updated Data Availability statement: “PAP-QMNIST data are available from the Zenodo repository, 10.5281/zenodo.7020311. The oral cancer data cannot be shared publicly because of the ethical permit restrictions. The values used to build graphs, the values behind the means, standard deviations, and other measures reported, as well as the weights for neural network models trained on PAP-QMNIST and oral cancer data, are provided in repository XXX.”

8. Your ethics statement should only appear in the Methods section of your manuscript. If your ethics statement is written in any section besides the Methods, please move it to the Methods section and delete it from any other section. Please ensure that your ethics statement is included in your manuscript, as the ethics statement entered into the online submission form will not be published alongside your manuscript.

Authors’ response: We have inserted the ethics statement as a subsection of the Methods section called “Ethical approval”.

Reviewer #1:

Comments:

1) In the abstract, you mention "This dataset serves as a proxy...". This needs to be better explained and describe what you aim to achieve and how you use it.

Authors’ response: We added a clarification in the abstract and “Data/PAP-QMNIST Dataset” subsection of the manuscript.

2) Have the authors considered self-supervised methods? Usually there are improvements when combined with MIL approaches. Some results even with simple methods would benefit the paper.

Authors’ response: SSL methods typically underperform more supervised AI approaches. In the study, we are applying end-to-end approaches because they are believed to learn better image representations as they are trained for the task of interest in contrast to image representations learned in an un/selfsupervised, not task-specific way. Even if incorporating SSL-based pretraining (thereby adding complexity to the paper), we see no reason to expect that this would change the relative performance of SIL and MIL approaches.

3) line 58 "comprise" should be "include"

Authors’ response: The word ‘comprise’ is now replaced.

4) Phrase in lines 64-68 should be rewritten to convery the meaning in a more clear manner.

Authors’ response: We rephrased and changed the order of the sentences in lines 64-68.

5) In lines 82-83 you write: "A main advantage of PAP-QMNIST is that is offers access to reliable GT annotation at the instance (cell) level". In the previous sentence, you write that GT annotation is not feasible. How both are true? Giving a brief overview of the dataset you created and what you aim to achieve would be beneficial here. Moreover, being more "precise" in the language would avoid confusion (it seems that PAP-QMNIST does not have cells at all).

Authors’ response: We agree and we added clarification to the text.

6) In the end of Introduction, I believe "splitting" the contributions and findings, as well as adding a small overview of your findings would be beneficial for the reader (to have a first impression on what to expect from the rest of the paper).

Authors’ response: We separated the contributions and findings.

7) Is there a reason for using F1 score instead of metrics like FAR, FRR? These metrics are interpreted in a manner that is easier to associate a cost for each action. An input of a medical expert, e.g. we are interested in very low number of False Negatives, would also add value to your findings.

Authors’ response: The SIL model with the highest F1 score in the window with the best moving average F1 score is chosen for evaluation at inference. F1 score is calculated based on weak labels, i.e., instances are given labels of a bag they belong to. F1 score is chosen as a typical measure in case of class imbalance; in our case, imbalance is in the number of instances with positive weak label and negative weak label. The F1 score, being a balanced combination of precision and recall, is used when there is a need for a single scalar to optimize. Since the instance GT is not available for training, but only weak labels, the use of a more tailored metric is not justified and could emphasize, for example, negative instances from positive bags more than negative instances from negative bags. Please note that we use different metrics for evaluation, such as accuracy at the bag level and Precision@Ki at the instance level. We noticed that the name Recall@Ki for the instance metric was misleading and changed it to fit the metric's purpose, which is to quantify the number of real key instances among the top K instances of the bag i (assuming that instances with top K weights/scores are predicted as positive). We insert a clarification for the choice of accuracy at the bag level and Precision@Ki at the instance level in the subsection “Experimental setup/Quantitative evaluation” of the manuscript.

8) In section "Considerd MIL method" some description/refresher would help the flow of the paper. For example, what is the modified version described in [13] (again a short description and the reader can refer to [13] for more details)? But throught the paper there isn't any model description.

Authors’ response: We made a clarification about the version of ABMIL we use. We included general diagrams of SIL and MIL models in the introduction and added models' details in the Supporting information section.

9) In line 177 you mention 24 patients, and in lines 185-186 you give some extra information. 

---

## [Decision Letter · Decision Letter 1]

9 Feb 2024

PONE-D-23-20689R1Deep multiple instance learning versus conventional deep single instance learning for interpretable oral cancer detectionPLOS ONE

Dear Dr. Koriakina,

Thank you for submitting your manuscript to PLOS ONE. After careful consideration, we feel that it has merit but does not fully meet PLOS ONE’s publication criteria as it currently stands. Therefore, we invite you to submit a revised version of the manuscript that addresses the points raised during the review process.

We look forward to receiving your revised manuscript.

Kind regards,

Sathishkumar Veerappampalayam Easwaramoorthy

Academic Editor

PLOS ONE

Journal Requirements:

Reviewers' comments:

Reviewer's Responses to Questions

**Comments to the Author**

1. If the authors have adequately addressed your comments raised in a previous round of review and you feel that this manuscript is now acceptable for publication, you may indicate that here to bypass the “Comments to the Author” section, enter your conflict of interest statement in the “Confidential to Editor” section, and submit your "Accept" recommendation.

Reviewer #2: All comments have been addressed

Reviewer #3: All comments have been addressed

2. Is the manuscript technically sound, and do the data support the conclusions?

Reviewer #2: Partly

Reviewer #3: Yes

3. Has the statistical analysis been performed appropriately and rigorously? 

Reviewer #2: Yes

Reviewer #3: Yes

4. Have the authors made all data underlying the findings in their manuscript fully available?

Reviewer #2: Yes

Reviewer #3: Yes

5. Is the manuscript presented in an intelligible fashion and written in standard English?

Reviewer #2: Yes

Reviewer #3: Yes

6. Review Comments to the Author

Reviewer #2: Dear authors,

I read with great interest the manuscript, which falls within the aim of this Journal. In my honest opinion, the topic is interesting enough to attract the readers’ attention. Nevertheless, authors should clarify some points and improve the discussion, as suggested below. Authors should consider the following recommendations:

In my opinion you have to improve the paper refering in the text how its really impoerant to refer and compare to the PAPILLOMAVIRUS and oral cancer correlation and how artifical ointelligence in this denatality ERA can be really importnat especially in pts with cervical and other malignanicues as endometrial cancer that need to preserv their fertility by oocite vitrification for future use before fertility sparing surgery treatment.

Tthe artifical intelligence contribution as well how is suggested to these pts to perform a NIPT TEST at beginning of pregnancy as well to follow up the impart of ASSITSTED REPRODUCTIVE TECHNOLOGY ART) in the newborn.

I SUGGEST YOU TO READ AND CITE THESE ARTICLES:

Open vs. closed vitrification system: which one is safer?

Endometrial Cancer in Reproductive Age: Fertility-Sparing Approach and Reproductive Outcomes

Neoadjuvant chemotherapy in advanced-stage ovarian cancer – state of the art

Fertility-Sparing Strategies for Early-Stage Endometrial Cancer: Stepping towards Precision Medicine Based on the Molecular Fingerprint

Neonatal Outcomes and Long-Term Follow-Up of Children Born from Frozen Embryo, a Narrative Review of Latest Research Findings

Circulating miRNAs as a Tool for Early Diagnosis of Endometrial Cancer-Implications for the Fertility-Sparing Process: Clinical, Biological, and Legal Aspects

Fresh vs. frozen embryo transfer in assisted reproductive techniques: a single center retrospective cohort study and ethical-legal implications

The Future Is Coming: Artificial Intelligence in the Treatment of Infertility Could Improve Assisted Reproduction Outcomes-The Value of Regulatory Frameworks

Cell-Free Fetal DNA and Non-Invasive Prenatal Diagnosis of Chromosomopathies and Pediatric Monogenic Diseases: A Critical Appraisal and Medicolegal Remarks

Impact of assisted reproduction techniques on the neuro-psycho-motor outcome of newborns: a critical appraisal

Sentinel Lymph Node Staging in Early-Stage Cervical Cancer: A Comprehensive Review

Reviewer #3: Accept

All comments have been addressed

I would like to thank the authors for their efforts

the conclusion is justified by the methods and the results

7. PLOS authors have the option to publish the peer review history of their article (what does this mean?). If published, this will include your full peer review and any attached files.

Reviewer #2: **Yes: **giuseppe gullo

Reviewer #3: No

---

## [Author Response · Author response to Decision Letter 1]

24 Mar 2024

We thank the reviewers and academic editor for providing suggestions on improving our publication. 

Authors’ response: We updated one citation from 

[1] Zhou H, Chen H, Yu B, Pang S, Cong X, Cong L. An end-to-end weakly supervised learning framework for cancer subtype classification using histopathological slides. Expert Systems with Applications. 2023; p. 121379. 

to 

[2] Zhou H, Chen H, Yu B, Pang S, Cong X, Cong L. An end-to-end weakly supervised learning framework for cancer subtype classification using histopathological slides. Expert Systems with Applications. 2024;237:121379. 

due to the replacement of a pre-print version of the publication by a version with a publisher’s standard format.

Reviewer # 2:

1. Dear authors,

I read with great interest the manuscript, which falls within the aim of this Journal. In my honest opinion, the topic is interesting enough to attract the readers’ attention. Nevertheless, authors should clarify some points and improve the discussion, as suggested below. Authors should consider the following recommendations:

In my opinion you have to improve the paper refering in the text how its really impoerant to refer and compare to the PAPILLOMAVIRUS and oral cancer correlation and how artifical ointelligence in this denatality ERA can be really importnat especially in pts with cervical and other malignanicues as endometrial cancer that need to preserv their fertility by oocite vitrification for future use before fertility sparing surgery treatment.

Tthe artifical intelligence contribution as well how is suggested to these pts to perform a NIPT TEST at beginning of pregnancy as well to follow up the impart of ASSITSTED REPRODUCTIVE TECHNOLOGY ART) in the newborn.

I SUGGEST YOU TO READ AND CITE THESE ARTICLES:

[3] Gullo G, Perino A, Cucinella G. Open vs. closed vitrification system: which one is safer?. European review for medical and pharmacological sciences. 2022 Feb;26(4):1065-7.

[4] Mutlu L, Manavella DD, Gullo G, McNamara B, Santin AD, Patrizio P. Endometrial cancer in reproductive age: fertility-sparing approach and reproductive outcomes. Cancers. 2022 Oct 22;14(21):5187.

[5] Margioula-Siarkou C, Petousis S, Papanikolaou A, Gullo G, Margioula-Siarkou G, Laganà AS, Dinas K, Guyon F. Neoadjuvant chemotherapy in advanced-stage ovarian cancer–state of the art. Menopause Review/Przegląd Menopauzalny. 2022 Dec 30;21(4):272-5.

[6] Gullo G, Cucinella G, Chiantera V, Dellino M, Cascardi E, Török P, Herman T, Garzon S, Uccella S, Laganà AS. Fertility-Sparing Strategies for Early-Stage Endometrial Cancer: Stepping towards Precision Medicine Based on the Molecular Fingerprint. International Journal of Molecular Sciences. 2023 Jan 3;24(1):811.

[7] Gullo G, Scaglione M, Cucinella G, Chiantera V, Perino A, Greco ME, Laganà AS, Marinelli E, Basile G, Zaami S. Neonatal outcomes and long-term follow-up of children born from frozen embryo, a narrative review of latest research findings. Medicina. 2022 Sep 4;58(9):1218.

[8] Piergentili R, Gullo G, Basile G, Gulia C, Porrello A, Cucinella G, Marinelli E, Zaami S. Circulating miRNAs as a Tool for Early Diagnosis of Endometrial Cancer–Implications for the Fertility-sparing Process: Clinical, Biological and Legal Aspects.

[9] Gullo G, Basile G, Cucinella G, Greco ME, Perino A, Chiantera V, Marinelli S. Fresh vs. frozen embryo transfer in assisted reproductive techniques: a single center retrospective cohort study and ethical-legal implications. European Review for Medical & Pharmacological Sciences. 2023 Jul 15;27(14).

[10] Medenica S, Zivanovic D, Batkoska L, Marinelli S, Basile G, Perino A, Cucinella G, Gullo G, Zaami S. The Future Is Coming: Artificial Intelligence in the Treatment of Infertility Could Improve Assisted Reproduction Outcomes—The Value of Regulatory Frameworks. Diagnostics. 2022 Nov 28;12(12):2979.

[11] Gullo G, Scaglione M, Buzzaccarini G, Laganà AS, Basile G, Chiantera V, Cucinella G, Zaami S. Cell-free fetal DNA and non-invasive prenatal diagnosis of chromosomopathies and pediatric monogenic diseases: A critical appraisal and medicolegal remarks. Journal of Personalized Medicine. 2022 Dec 20;13(1):1.

[12] Gullo G, Scaglione M, Cucinella G, Perino A, Chiantera V, D’Anna R, Laganà AS, Buzzaccarini G. Impact of assisted reproduction techniques on the neuro-psycho-motor outcome of newborns: a critical appraisal. Journal of Obstetrics and Gynaecology. 2022 Oct 3;42(7):2583-7.

Authors’ response: 

We would like to note to the Reviewer #2 and the editor that we have already cited one of the by Reviewer #2 proposed publications,

[10] Medenica S, Zivanovic D, Batkoska L, Marinelli S, Basile G, Perino A, Cucinella G, Gullo G, Zaami S. The Future Is Coming: Artificial Intelligence in the Treatment of Infertility Could Improve Assisted Reproduction Outcomes—The Value of Regulatory Frameworks. Diagnostics. 2022 Nov 28;12(12):2979., 

referring to it in our section ‘Limitations and future directions’. 

After a careful consideration of the remaining publications in the proposed literature list, we came to the conclusion that these publications are not relevant enough to the topic and goals of our publication; the insertion of information about Assisted reproductive technology art could be misleading for potential readers: we prefer to keep it clear that this paper presents a study on weakly-supervised DL methods applied to oral cancer classification. However, we are grateful for the feedback.

Reviewer #3:

1. Accept

All comments have been addressed

I would like to thank the authors for their efforts

the conclusion is justified by the methods and the results

Authors’ response: We thank Reviewer #3 for the valuable feedback that improved the paper and for the kind words.

Additional updates:

After a careful self-review, we observed that the way in which we computed the threshold for predicting positive versus negative test bags using SIL by averaging validation thresholds over all the folds could be questioned. The reason for that is that test bags in one fold may appear as train or validation bags of another fold. To make the separation of test and training data unquestionable, we decided to predict bags in a different way, by computing thresholds for each fold separately, without any averaging. Namely, the threshold tf for each fold f, separating positive and negative test bags, is chosen as the middle value between the fifth percentile (calculated using linear interpolation) of the percentage of instances classified as positive (softmax score>0.5) in positive and negative bags of fold f from train and validation sets. The update has a minor effect on the average and standard deviation of bag accuracy and precision at the instance level, however, it strengthens the overall conclusions of the paper. We have updated the figures and text of our paper correspondingly, with all changes clearly indicated. We have also corrected a few remaining typos and minor linguistic issues.

---

## [Editor Report · Decision Letter 2]

28 Mar 2024

Deep multiple instance learning versus conventional deep single instance learning for interpretable oral cancer detection

PONE-D-23-20689R2

Dear Dr. Koriakina,

We’re pleased to inform you that your manuscript has been judged scientifically suitable for publication and will be formally accepted for publication once it meets all outstanding technical requirements.

Kind regards,

Sathishkumar Veerappampalayam Easwaramoorthy

Academic Editor

PLOS ONE